# Response of Arctic mixed-phase clouds to aerosol perturbations under different surface forcings

Gesa K. Eirund[1], Anna Possner[2*], and Ulrike Lohmann[1]

[1]Institute for Atmospheric and Climate Sciences, ETH Zurich, Zurich, Switzerland
[2]Department of Global Ecology, Carnegie Institution for Science, Stanford, California, USA
[*]Now at: Institute for Atmospheric and Environmental Sciences, Goethe-University Frankfurt, Frankfurt, Germany

**Correspondence:** Gesa K. Eirund (gesa.eirund@env.ethz.ch)

**Abstract.** The formation and persistence of low lying mixed-phase clouds (MPCs) in the Arctic depends on a multitude of processes, such as surface conditions, the environmental state, air mass advection and the ambient aerosol concentration. In this study, we focus on the relative importance of different instantaneous aerosol perturbations (cloud condensation nuclei and ice nucleating particles; CCN and INPs, respectively) on MPC properties in the European Arctic. To address this topic, we performed high resolution large eddy simulations (LES) using the Consortium for Small-scale Modeling (COSMO) model and designed a case study for the Aerosol-Cloud Coupling and Climate Interactions in the Arctic (ACCACIA) campaign in March 2013. Motivated by ongoing sea ice retreat, we performed all sensitivity studies over open ocean and sea ice to investigate the effect of changing surface conditions. We find that surface conditions highly impact cloud dynamics, consistent with the ACCACIA observations: over sea ice, a rather homogeneous, optically thin, mixed-phase stratus cloud forms. In contrast, the MPC over the open ocean has a stratocumulus-like cloud structure. With cumuli feeding moisture into the stratus layer, the cloud over the open ocean features a higher liquid (LWP) and ice water path (IWP) and has a lifted cloud base and cloud top compared to the cloud over sea ice.

Furthermore, we analyzed the aerosol impact on the sea ice and open ocean cloud regime. Perturbation aerosol concentrations relevant for CCN activation were increased to a range between 100 to 1000 $\mathrm{cm}^{-3}$ and INP perturbations were increased by 100% and 300% as compared to the background concentration (at every grid point and at all levels). The perturbations are prognostic to allow for fully interactive aerosol-cloud interactions. Perturbations in the INP concentration increase IWP and decrease LWP consistently in both regimes. The cloud microphysical response to potential CCN perturbations occurs faster in the stratocumulus regime over the ocean, where the increased moisture flux favors rapid cloud droplet formation and growth, leading to an increase in LWP following the aerosol injection. In addition, IWP increases through new ice crystal formation by increased immersion freezing, cloud top rise, as well as subsequent growth by deposition. Over sea ice, the maximum response in LWP and IWP is delayed and weakened compared to the response over the open ocean surface.

Additionally, we find the long-term response to aerosol perturbations being highly dependent on the cloud regime. Over the open ocean, LWP perturbations are efficiently buffered after 18 h simulation time. Increased ice and precipitation formation relax the LWP back to its unperturbed range. On the contrary, over sea ice the cloud evolution remains substantially perturbed with CCN perturbations ranging from 200 to 1000 CCN $\mathrm{cm}^{-3}$.

# 1 Introduction

Clouds play a crucial role in the hydrological cycle and the radiative balance of the Earth-atmosphere system. However, clouds still comprise high uncertainties and their behavior under climate change scenarios is not yet well-understood. Hence, the magnitude of the cloud radiative forcing in the upcoming years remains unclear (IPCC, 2013). Mixed-phase clouds (MPCs)
contain both phases, i.e. ice and water, and are important for the radiative balance (Lohmann, 2002) and climate sensitivity (Tan et al., 2016). MPCs occur either in regions of deep convection, where the cloud top reaches temperatures low enough for ice formation (Rosenfeld and Woodley, 2000), in mountainous terrain (Lloyd et al., 2015a; Farrington et al., 2016; Lohmann et al., 2016), or in cold regions of the planet, i.e. in high latitudes (Morrison et al., 2011). In the Arctic, MPCs occur approximately 40% of the time (Shupe et al., 2006) and are often observed as persistent low clouds (Shupe et al., 2011). Their radiative forcing
at the surface is still ambiguous and determined in part by the distinct seasonal cycle at high latitudes. In summer, the reflection of incoming shortwave (SW) radiation dominates, while during the rest of the year absorption and emission of longwave (LW) radiation prevails, causing a warming at the surface (Curry et al., 1996). In recent decades the Arctic has been warming at a faster rate than the rest of the globe (Serreze and Barry, 2011). As changes in the Arctic can impact mid-latitude weather conditions, the climate state of the Arctic is important not only regionally but also hemisphere-wide (Cohen et al., 2014; Ye
et al., 2018). Due to their strong radiative impact, MPCs can alter the Arctic climate system (e.g. Bennartz et al., 2013; Van Tricht et al., 2016), potentially accelerating or slowing the current high latitude warming.

Arctic MPC fraction and phase partitioning are governed by a multitude of processes operating in conjunction across a wide range of spatial scales; such as the large-scale dynamical forcing, surface processes, as well as the ambient aerosol concentration. The large-scale dynamical forcing determines air mass and hence water vapor advection, which is found to be
crucial for the persistence of Arctic MPCs (Morrison et al., 2011; Sedlar et al., 2012; Loewe et al., 2017). With ongoing sea ice loss and the possibility of an ice-free Arctic by mid century (Overland and Wang, 2013), the impact of surface conditions on Arctic MPCs has gained increasing attention in the past decade (e.g. Schweiger et al., 2008; Palm et al., 2010; Vavrus et al., 2010; Liu et al., 2012; Sotiropoulou et al., 2016; Young et al., 2016). A more exposed open ocean surface has potential implications for cloud dynamics (Schweiger et al., 2008; Sotiropoulou et al., 2016; Young et al., 2016, 2018). Schweiger et al.
(2008) using the 40-yr European Centre for Medium-Range Weather Forecasts Re-Analysis (ERA-40) product demonstrated that sea ice loss increased boundary layer height and led to more midlevel clouds. In addition, Sotiropoulou et al. (2016) found increased stratocumulus or cumulus cloud formation over the ocean in contrast to thin stratus clouds over sea ice in observations from the Arctic Clouds in Summer Experiment (ACSE) campaign. These observed changes in cloud height were also observed during the Aerosol-Cloud Coupling And Climate Interactions in the Arctic (ACCACIA) campaign (Young et al.,
2016). Besides, the authors reported fewer and larger cloud droplets as well as increased precipitation rates over the open ocean compared to over sea ice. In LES simulations for the same case, Young et al. (2017) could reproduce these observations and Young et al. (2018) simulated cumuli tower development over a warming open ocean surface, in agreement with previous results of more convective cloud systems over a destabilized surface.

In addition to increased surface heat fluxes, aerosol emissions may increase in the Arctic (Struthers et al., 2011; Browse et al., 2014; Gilgen et al., 2018; Stephenson et al., 2018) which could impact cloud microphysics. Since the Arctic is a pristine environment and aerosol concentrations are generally lower than in the lower and mid-latitudes (Moore et al., 2013; Schmale et al., 2018), any aerosol perturbations could substantially impact MPC formation and persistence. With decreasing sea ice, trans-Arctic shipping is also projected to increase, exerting local aerosol perturbations (Hobbs et al., 2000; Khon et al., 2010; Peters et al., 2011). An increased availability of cloud condensation nuclei (CCN) resulting from both sea salt and dimethyl sulfide emissions from the ocean and predicted ship emissions may lead to increased cloud formation and a net surface cooling during summer, as projected by global climate and earth system models (Gilgen et al., 2018; Stephenson et al., 2018). Locally, aerosols released in ship tracks alone can change cloud liquid and ice water content (LWC and IWC, respectively) as found in studies of Christensen et al. (2014) and Possner et al. (2017). Equivalently, a reduction in the ambient CCN and hence cloud droplet number concentration ($N_{drop}$) could induce cloud dissipation (Mauritsen et al., 2011; Loewe et al., 2017; Stevens et al., 2018). However, disentangling the competing effects of environmental conditions and aerosol disturbances appears challenging (Jackson et al., 2012). In the past, Stevens and Feingold (2009) argued for a buffered aerosol response in certain cloud regimes. For mid-latitude convective clouds, Miltenberger et al. (2018) showed that cloud fraction is not impacted by aerosol perturbations, but that aerosols may affect the organization of cloud pockets with fewer, but larger cloud cells under levels of increased pollution. In simulations of trade wind shallow cumuli by Seifert et al. (2015) an initial aerosol response is seen, with an increased number of cumulus structures and decreased precipitation. Yet the system efficiently returns to an organized cloud structure in a quasi-stationary state after some hours, which is insensitive to the background aerosol concentration. Turbulent mixing, entrainment and detrainment of aerosols out of polluted regions could potentially also impact the aerosol concentration and the long-term aerosol response, as has been simulated by Berner et al. (2015) in large eddy simulations (LES) of ship tracks in the Monterey Bay. On the other hand, Igel et al. (2017) found that entrainment of aerosols from the free troposphere into the boundary layer represents an important source of aerosol particles for Arctic MPCs as the authors showed in observations and LES simulations of the Arctic Summer Cloud Ocean Study (ASCOS) field campaign.

In this study, we investigate how the response to increased aerosol concentrations may differ for different cloud regimes of Arctic MPCs. For this purpose we perform high-resolution idealized LES to resolve the multitude of boundary layer processes that impact the cloud state. We contrast our results for different surface conditions (open ocean surface versus sea ice) and apply different perturbations across a $\pm 2$ K temperature range. To validate our simulations we use observations obtained during the recent ACCACIA campaign (Lloyd et al., 2015b; Young et al., 2016) in the European Arctic.

## 2   Model description and setup

LES are performed with the Consortium for Small-scale Modeling (COSMO) model in its configuration for idealized LES experiments (Schättler et al., 2000). COSMO-LES has been proven to simulate MPCs in the Arctic with reasonable accuracy (Possner et al., 2017). Here, we simulate a single-layer stratocumulus case during the ACCACIA campaign on March $23^{rd}$, 2013. All simulations are initialized with the dropsonde profile number 5 released during the campaign (Young et al., 2016).

The obtained profiles are smoothed to exclude small-scale variability from the measurements as model input. In addition, the water vapor mixing ratio ($q_v$) was increased by 20% to account for the dry bias in dropsonde data (Ralph et al., 2005; Young et al., 2016). Note that in contrast to Young et al. (2017) we initialize the open ocean as well as the sea ice simulations with the same atmospheric profile, to narrow down dynamic changes in the cloud-topped boundary layer to changed surface conditions alone (i.e. turbulent surface fluxes) and exclude any impact from varying large-scale conditions or boundary layer stability.

The domain covers a 19.2 km x 19.2 km large area centered around the location of the release of dropsonde number 5 (75°N, 24.5°E). The horizontal resolution is 120 m, the vertical resolution is variable and specified with 20 to 25 m within the entire boundary layer and coarser resolution above cloud top up to the model top at 23 km. The temporal resolution is 2 s and the model has been run for 20 h, including a 1.5 h spin-up period. Radiation is treated interactively according to the Ritter and Geleyn (1992) radiation scheme and includes a diurnal cycle. The cloud microphysical tendencies are parameterized following the Seifert and Beheng (2006) two-moment scheme. The scheme considers five hydrometeor types (cloud droplets, rain drops, cloud ice, snow and graupel) represented as gamma distributions with prescribed shape parameters and prognosed bulk mass and number concentrations. As in Possner et al. (2017) we use a prognostic treatment of ice nucleating particles (INPs) while we keep the background CCN fixed, with cloud droplet activation calculated according to Koehler theory (Nenes and Seinfeld, 2003). The fixed background CCN ensure that sufficient CCN are available throughout the whole simulation for droplet activation. CCN are assumed to be pure ammonium bisulfate particles. Prognostic INPs are implemented as in Solomon et al. (2015). The scheme parameterizes immersion freezing following the DeMott et al. (2015) temperature dependence and captures the depletion and replenishment of INPs. Following the COSMO setup for the model intercomparison performed by Stevens et al. (2018), ice crystals and snow flakes are assumed to be dendrites. As secondary ice processes are observationally poorly constrained, only the HP-mechanism (Hallett and Mossop, 1974) is included in our model, which is inefficient at cold temperatures (-15 to -20°C).

We initialize the simulations with one background mode of potential CCN (0.2 μm mean diameter and 1.5 standard deviation), represented by a lognormal size distribution. For direct comparison to observations and the Young et al. (2017) model study, the CCN concentrations were chosen to match the observed $N_{drop}$ over the ocean (Young et al., 2016) and the fixed $N_{drop}$ in Young et al. (2017), and were set to 100 cm$^{-3}$. As we do not expect every CCN to activate, we initialized with a CCN concentration larger than the mean $N_{drop}$ measured over the ocean. The initialized CCN concentration is still within the spread of the measured $N_{drop}$ range though. INPs were initialized with a concentration of 3.3 L$^{-1}$, which is at the high end of predicted ice crystal number concentrations ($N_{ice}$) by different parameterizations in Young et al. (2016) (assuming one INP per ice crystal). Due to the interactive INPs in our simulations, we used a relatively high initial INP concentration to prevent an underestimation of $N_{ice}$. For simplicity we assumed a constant aerosol profile with height. As for the background thermodynamic conditions, we kept the background aerosol concentrations the same in the open ocean and sea ice case.

We performed control simulations over sea ice and open ocean and evaluated these against available observations. For the sea ice case, the COSMO sea ice model (Mironov et al., 2012) was switched on. To exclude influences from variable turbulent fluxes, the sensible and latent heat flux were set to 25 and 23 W m$^{-2}$ over ocean and to 1 and 0.8 W m$^{-2}$ over sea ice. These prescribed fluxes are at the lower end of the observed range (Young et al., 2016). However, larger fluxes were found to

increase the strength and size of the convective cells in sensitivity simulations not shown here. Therefore, we would need larger domain sizes to simulate cases with larger surface fluxes. This was not possible due to the high computational demand of each simulation. Surface roughness length was assumed to be higher over the ocean with 0.0002 m in contrast to 0.0001 m over sea ice. Divergence was prescribed as zero at the surface and was relaxed linearly to $4*10^{-6}$ s$^{-1}$ at the inversion height and kept constant above. To compensate for the subsidence heating, we included negative horizontal advective temperature tendencies, while all other tendencies were set to zero to prevent any influence of boundary layer moistening or drying by large-scale advection.

## 2.1 Setup of perturbation experiments

In order to study the effects of aerosol perturbations, an additional, fully prognostic mode of potential CCN or INPs was released at every grid point at every height after 1.5 h of simulation time, i.e. following the initial surface precipitation peak. At this time step, the full aerosol perturbation was released. The perturbation mode was assumed to have the same chemical composition but to be at a slightly smaller size than the background mode (0.19 $\mu$m). The smaller size ensures the perturbation mode to activate later than the background mode according to its implementation in the aerosol scheme. Both aerosol perturbations are prognostic, meaning that aerosols are advected throughout the domain, are depleted by cloud droplet or ice crystal formation and precipitation, and are released back into the atmosphere through evaporation or sublimation.

Perturbation aerosol concentrations relevant for CCN activation were increased by 100, 200, 500, and 1000 cm$^{-3}$. For INP perturbations we once perturbed with the background concentration (3.3 L$^{-1}$ for a temperature range of 250.5-258 K) and once increased the initial INP concentration by a factor of 3 (10 L$^{-1}$). A summary of all performed simulations can be found in Table 1.

Given the pronounced sensitivity of high-latitude cloud processes to atmospheric temperature (e.g. Devasthale and Thomas, 2012), we test the robustness of our results across a $\pm 2$ K temperature change of the background state. In these experiments the entire initial temperature profile was shifted towards colder or warmer temperatures at constant relative humidity.

| Name | CCN perturb ($\mathrm{cm}^{-3}$) | INP perturb ($\mathrm{L}^{-1}$) | T perturb (K) |
|---|---|---|---|
| *ocean_/ice_control* | - | - | - |
| *ocean_/ice_100CCN* | 100 | - | - |
| *ocean_/ice_200CCN* | 200 | - | - |
| *ocean_/ice_500CCN* | 500 | - | - |
| *ocean_/ice_1000CCN* | 1000 | - | - |
| *ocean_/ice_3INP* | - | 3 | - |
| *ocean_/ice_10INP* | - | 10 | - |
| *ocean_/ice_control+2K* | - | - | +2 |
| *ocean_/ice_control-2K* | - | - | -2 |
| *ocean_/ice_1000CCN+2K* | 1000 | - | +2 |
| *ocean_/ice_1000CCN-2K* | 1000 | - | -2 |
| *ocean_/ice_10INP+2K* | - | 10 | +2 |
| *ocean_/ice_10INP-2K* | - | 10 | -2 |

**Table 1.** Summary of all experiments performed. In all simulations the fixed background CCN concentration is $100\,\mathrm{cm}^{-3}$ and the prognostic INP concentration is set to 3.3 INP $\mathrm{L}^{-1}$. All settings listed here were run over open ocean and sea ice surface.

## 3 Evaluation of background state

The local atmospheric conditions over open ocean as observed during the ACCACIA campaign (hereafter named *observations*) are characterized by a single temperature inversion at 1.3 km, capping a single-layer MPC between approximately 0.3 and 1.2 km (Young et al., 2016). Our simulated case similarly features a strong inversion ($\Delta\theta =6$ K) at a height of 1.4 km, capping a single cloud layer below (Fig. 1). The boundary layer in both *control* simulations (named *ocean_control* and *ice_control* over open ocean and sea ice, respectively) is stably stratified, as seen in the positive gradient in the ice-liquid potential temperature

($\theta_{il}$) and the negative gradient in the total water content ($q_t$) in Fig. 1. Over the ocean surface an unstable surface layer forms due to the non-zero surface fluxes. The remainder of the boundary layer is stably stratified, which prevents the formation of a well-mixed boundary layer. As a result of stronger surface fluxes, the boundary layer retains more water vapor over the ocean as compared to sea ice (Fig. 1a,b).

    Our model successfully simulates a liquid-topped MPC with ice sedimenting out of the liquid layer in both *control* simula-

tions, in agreement with *observations*. The observed cloud properties obtained from Young et al. (2016), our simulated values of the unperturbed simulations, as well as the LES results from Young et al. (2017) are summarized for comparison in Table 2. From Young et al. (2017) we only included the simulation using the ice parameterization that was fitted to the *observations* (termed ACC), which best reproduced the observed case (Young et al., 2017).

    The simulated mean $N_{ice}$ of 0.27 $\mathrm{L}^{-1}$ in *ocean_control* (Table 2) is slightly lower compared to *observations*, but within the

observed range. $N_{drop}$ agrees well in our model simulations as compared to *observations*, but the maximum $N_{drop}$ in Figure

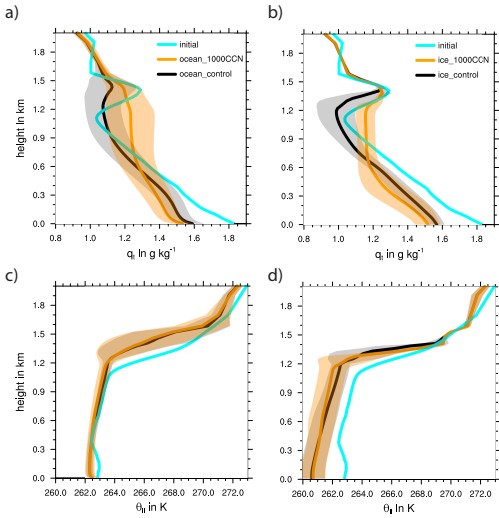

**Figure 1.** Time and domain averaged ($\pm$1 standard deviation) a) total water content $q_t$ ($q_t = q_c + q_v + q_i$) in the *ocean_control* simulations as well as most perturbed *ocean_1000CCN* simulation, b) total water content $q_t$ in the *ice_control* and *ice_1000CCN* simulation, c) ice-liquid potential temperature ($\theta_{il}$) in the *ocean_control* and *ocean_1000CCN* simulation and d) $\theta_{il}$ in the *ice_control* and *ice_1000CCN* simulation (for an overview of the simulations refer to Table 1). The blue lines represent the modeled initial values (i.e. timestep zero).

2a is simulated at a higher altitude (1.4 km instead of 1.0 km) due to the upward shift of the simulated stratiform cloud deck. The cloud droplet radius ($R_{drop}$) is smaller than observed, due to an underestimation of the liquid water mixing ratio (LWMR) in the *ocean_control* simulation by a factor of 2. This underestimation of the liquid phase is a general issue in high-resolution simulations of mixed-phase clouds. In particular, the potential impact of the autoconversion rate on cloud evolution in a similar
context has recently been discussed in Stevens et al. (2018).

    The *ice_control* simulation can only be compared to *observations* in qualitative terms, as the initialization relies on the open ocean dropsonde profile (see section 2). In the observations, the boundary layer over the sea ice was less well-mixed and colder and drier compared to the open ocean (Young et al., 2016). As a result, the observed LWMR is smaller over sea ice than over the ocean, which is reproduced in our simulations. Our simulated $N_{ice}$ is also considerably lower over sea ice than over ocean.
In contrast to *observations*, $R_{drop}$ is only 0.7 µm smaller in *ice_control* than in *ocean_control*, instead of 5 µm smaller in the *observations*. Additionally, $N_{drop}$ is smaller instead of larger in *ice_control* (Table 2 and Fig. 2b). We relate these differences in cloud properties between our simulated and the observed MPC to the difference in the observed and simulated thermodynamic profiles: the drier boundary layer observed over sea ice suppresses cloud droplet growth. Moreover, the warmer and more turbulent boundary layer over the open ocean favors collision-coalescence of cloud droplets, leading to larger and fewer $N_{drop}$
over the open ocean. By choosing the same initial conditions for our open ocean and sea ice simulations, these processes are not equally represented.

**Table 2.** Averaged (±1 standard deviation) cloud properties derived from the ACCACIA in-situ *observations* (Young et al., 2016, 2017), the Young et al. (2017) LES simulations, and the *ocean_control* and *ice_control* simulations (as temporal means over 2-20 h). As in the observations, all modeled quantities represent in-cloud values (cloud liquid content $q_c$>0.01 g m$^{-3}$ for LWMR, $N_{drop}$, and $R_{drop}$ and cloud ice content $q_i$>0.001 g m$^{-3}$ for $N_{ice}$ and $R_{ice}$).

|  | LWMR (g kg$^{-1}$) | $N_{drop}$ (cm$^{-3}$) | $N_{ice}$ (L$^{-1}$) | $R_{drop}$ (µm) | $R_{ice}$ (µm) |
|---|---|---|---|---|---|
| *observations* ocean | 0.24±0.13 | 63±30 | 0.55±0.95 | 10 | - |
| *ocean_control* | 0.11±0.08 | 48±15 | 0.27±0.20 | 6.5±1.7 | 15.5±2.0 |
| Young et al. (2017) LES | 0.06 | 100 | 0.34 | 10 | 30 |
| *observations* sea ice | 0.05±0.04 | 110±36 | 0.47±0.86 | 5 | - |
| *ice_control* | 0.06±0.05 | 40±18 | 0.08±0.05 | 5.8±1.8 | 18.0±2.9 |

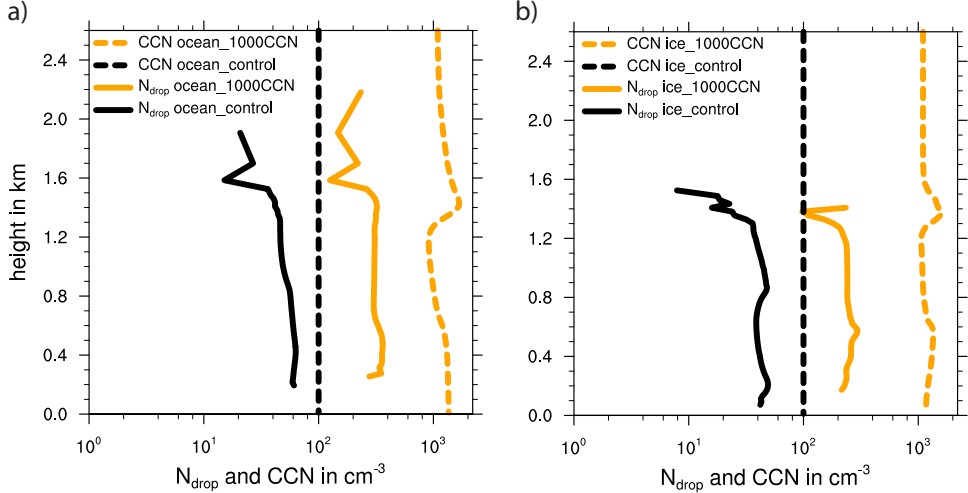

**Figure 2.** Average (2-20 h) $N_{drop}$ (solid lines) and the sum of all CCN tracers, i.e. background and perturbation mode, (dashed lines) in the a) *ocean_control* simulations as well as most perturbed *1000CCN* simulation and b) *ice_control* simulations as well as most perturbed *1000CCN* simulations.

## 4   Surface flux impact on cloud dynamics

The simulated effect of surface fluxes is illustrated in Fig. 3, showing a snapshot of the updraft velocities and LWP over ocean and sea ice after 3 h of simulation time. The different surface conditions lead to two different cloud regimes: over ocean, where surface fluxes are increased, the updrafts are higher, leading to cumulus towers detraining into the stratus deck and to a domain-wide shallow stratocumulus cloud structure. Within the shallow cumuli the LWP increases up to 300 g m$^{-2}$, 4 times

higher than in the surrounding stratus layer. In contrast, over sea ice the updrafts are low and a spatially homogeneous stratus forms. The LWP of the stratus cloud remains below $80\,\mathrm{g\,m^{-2}}$.

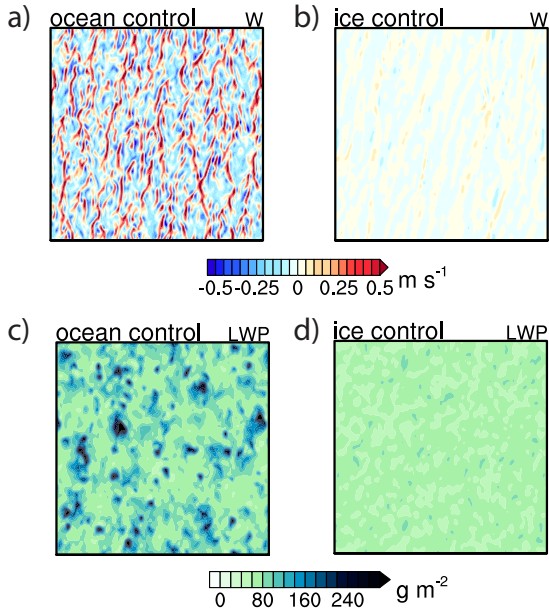

**Figure 3.** Snapshot at 3 h simulation time of a,b) sub-cloud updraft speed at 100 m and c,d) LWP for the *ocean_control* (left) and the *ice_control* case (right).

These dynamic differences feed back onto the vertical cloud structure (Fig. 4). Supported by the stronger updrafts over the

open ocean, the cloud base and top of the stratiform cloud deck are lifted by 200 m and 100 m, respectively, as compared to the cloud over sea ice (dashed lines in Fig. 4). These high updrafts over the open ocean sustain an increased rate of cloud droplet activation. Despite this increased rate of cloud droplet activation, the mean effective radius remains unchanged between the two cloud regimes due to the increased rate of condensate forming in the updraft. The higher updrafts also facilitate rain formation over the open ocean, where droplets can grow at a faster rate than in the surrounding stratus cloud. As a result,

total precipitation is increased over the open ocean (on average $1.12\,\mathrm{mm\,d^{-1}}$ as opposed to $0.59\,\mathrm{mm\,d^{-1}}$ above sea ice, Fig. S1a,b). Over sea ice, relatively low updraft speeds prohibit a strong upward moisture flux into the cloud layer due to the large thermodynamic stratification in the sub-cloud layer. This results in a drier boundary layer at cloud height and an optically thinner cloud (Table 3).

In addition to $N_{drop}$, $N_{ice}$ is also increased in *ocean_control* as compared to *ice_control*. As suggested by Garrett and Zhao (2006), the higher liquid water content in the air column increases the cloud LW emissivity. Thus, the higher LWP over the open ocean increases LW cloud top cooling, which initiates immersion freezing at cloud top (Fig. 5a,b). Through cooling in

the updrafts and more available moisture, ice crystals can grow more efficiently by vapor deposition over the ocean (Fig. 6a,b). Overall, these processes lead to a higher IWP over open ocean than over sea ice. Note that over the ocean sedimenting ice in

the form of snow contributes to 20% of total rain and snow at the surface, while over sea ice this is reduced to 2%.

These differences in cloud structure and properties (i.e. changes in cloud base and top, liquid and ice content, and precipitation efficiency) between the two cloud regimes agree with *observations* and previous LES results (Young et al., 2017).

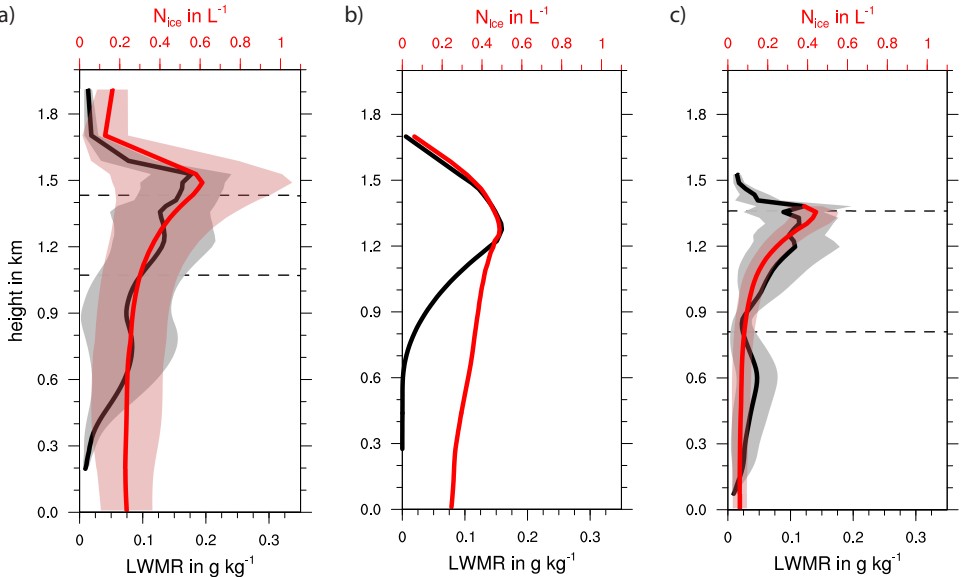

**Figure 4.** Domain and time averaged (2-20 h) $\pm$standard deviation of $N_{ice}$ (red) and cloud liquid water mixing ratio (black) in the a) *ocean_control*, b) Young et al. (2017) LES simulations and c) *ice_control* simulation. Only in-cloud values ($q_c$>0.01 g m$^{-3}$) are plotted. The horizontal dashed lines represent the modeled cloud base and cloud top, where 80% of the domain grid points are cloud-covered.

Due to the distinctly different cloud dynamics in both regimes, also the effect of the aerosol perturbations on the clouds differs. In the following we present results from several sensitivity simulations, where we investigated the cloud response to

CCN and INP perturbations across different temperature ranges for the two cloud regimes.

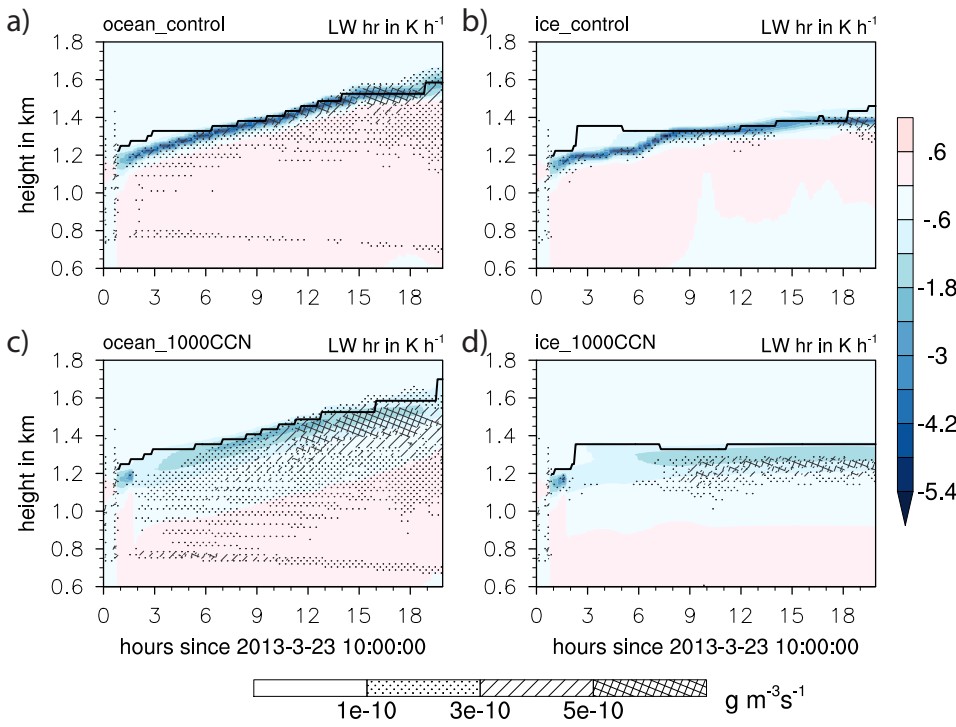

**Figure 5.** Domain averaged LW heating rate (color), immersion freezing rate (hatching) and cloud top of the uppermost cloud layer, where 80% of the domain grid points are cloud-covered ($q_c$>0.01 $\mathrm{g\,m^{-3}}$) are shown for the a) *ocean_control* and b) *ice_control* simulations and c,d) the respective *1000CCN* simulations. Only the range where immersion freezing occurs (T <258 K) is shown.

## 5 Robustness to perturbations in microphysics

### 5.1 Response to CCN perturbations

We performed simulations with potential CCN perturbations ranging from 100 to 1000 CCN $\mathrm{cm^{-3}}$. These number concentrations are higher than what would locally be expected from sea ice loss (Browse et al., 2014), but within the range of CCN
concentrations measured in ship exhaust plumes (Hobbs et al., 2000) or Arctic haze conditions in spring (Rogers et al., 2001). The perturbations were applied (as described in section 2) following the strong precipitation event 1.5 h after initialization. Over the ocean, the cloud responds almost immediately to CCN perturbations with an increase in LWP (Fig. 7a). A doubling of the initial CCN concentration (100 CCN $\mathrm{cm^{-3}}$) is sufficient to increase mean LWP by 13 $\mathrm{g\,m^{-2}}$ within the first hour after seeding. This equals a 13% change in LWP between *ocean_control* and *ocean_100CCN* and is comparable to the observed
LWP change in ship tracks by Christensen et al. (2014).

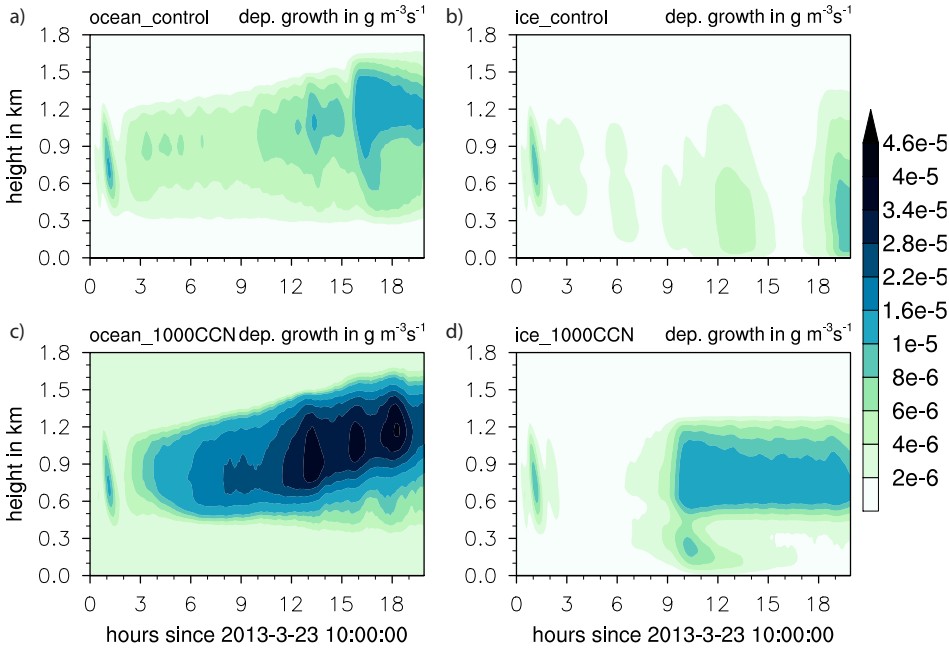

**Figure 6.** Domain averaged depositional growth rate for the a) *ocean_control* and b) *ice_control* simulations and c,d) the respective *1000CCN* simulations. Only the vertical range where immersion freezing occurs (T <258 K) is shown. Note the non-linear colorbar.

Elevated CCN concentrations in combination with strong updrafts allow fast additional droplet formation, which immediately increases the in-cloud vertical mean $N_{drop}$ from 49 to 201 $\mathrm{cm}^{-3}$ directly after seeding and decreases $R_{drop}$ from 6 to 4 μm in the *ocean_1000CCN* simulation (Fig. S2). This decrease in radius is expected according to the Twomey effect (Twomey, 1974). In addition, we also see a 20% increase in liquid water content through a delay of warm rain formation. Con-

sequently, with increasing CCN perturbation, LWP successively increases, however, a further increase in perturbation strength from 500 $\mathrm{cm}^{-3}$ to 1000 $\mathrm{cm}^{-3}$ induces only a slight increase in LWP. As the total water content is similar for *ocean_500CCN* and *ocean_1000CCN* (Fig. S3), the boundary layer seems to be saturated for a CCN perturbation of 500 $\mathrm{cm}^{-3}$. All available precipitation has been suppressed and further growth of the mixed-layer is inhibited for CCN perturbations >500 $\mathrm{cm}^{-3}$. Ad-ditionally, in these two most perturbed simulations, the cloud top rises and the cloud deepens through overshooting cumulus

towers that detrain moisture into the free troposphere and pre-condition the layers above cloud top for further cloud growth (Fig. 5a,c and S4). The cloud top rise in simulations perturbed by CCN could be a result of latent heat release during cloud droplet formation which feeds back onto the updraft velocities. For CCN perturbations below 200 $\mathrm{cm}^{-3}$, this additional latent heating

might not be enough to sustain further cloud growth and the cloud top does not rise in *ocean_100CCN* and *ocean_200CCN* as compared to *ocean_control*.

Apart from changes in LWP, also $N_{ice}$ and IWP are affected by CCN perturbations (Fig. 7c and Fig. S5a). Firstly, the cloud deepening in *ocean_500CCN* and *ocean_1000CCN* (Fig. S4) results in an increase in $N_{ice}$ in the respective simulations, as at higher altitudes new INPs can be entrained and become available for immersion freezing. Immersion freezing is also more efficient throughout the cloud, as the higher LWP radiatively cools the cloud layer over a larger area as compared to *ocean_control* (Fig. 5c), which additionally increases $N_{ice}$ in the perturbed simulations. Secondly, growth by vapor deposition is consider-

ably increased in the perturbed simulations (Fig. 6c). The radiative cooling in addition to slightly colder temperatures at cloud top through the cloud deepening create favorable conditions for ice crystal growth through the Wegener-Bergeron-Findeisen (WBF) process (Wegener, 1911; Bergeron, 1935; Findeisen, 1938). This cooling of the cloud-driven mixed layer together with higher $N_{ice}$ favor more efficient depositional growth in all CCN sensitivity simulations. Besides, the sub-cloud and cloudy layer become increasingly well-mixed and moistened with respect to *ocean_control* in all sensitivity simulations (Fig. 1 and

Fig. S3), such that the boundary layer remains supersaturated with respect to water and the liquid as well as the ice phase can grow simultaneously. In *ocean_500CCN* and *ocean_1000CCN* the stronger cloud top rise and cloud layer cooling sustain an immediate increase in the depositional growth rate, which increases IWP in these simulations as compared to *ocean_100CCN* and *ocean_200CCN* throughout the simulated time period. The importance of depositional growth in simulations perturbed by CCN agrees with recent results from Solomon et al. (2018).

The response to CCN perturbations strongly depends on the cloud regime. Due to the lower updrafts and the decreased vertical moisture transport over sea ice, the increase in $N_{drop}$ after the CCN injection is lower than over the ocean (Fig. S2). Limited by moisture transport into the cloud layer over sea ice, the increase in LWP is weaker than over the open ocean (Fig. 7b). However, the spatial variability of LWP is reduced over sea ice due to the more stratiform cloud deck. Therefore, smaller perturbations in LWP are considered outside the background variability in polluted simulations above sea ice. Indeed a CCN

perturbation of 100 $\mathrm{cm}^{-3}$ is sufficient above sea ice, while a perturbation of 200 $\mathrm{cm}^{-3}$ is needed above the ocean to induce LWP perturbations outside the simulated background conditions.

    Over sea ice, IWP and $N_{ice}$ reach a maximum shortly after the maximum increase in LWP (Fig. 7d and S5b). As over the ocean, LW cooling over a larger vertical range in the CCN perturbation simulations triggers immersion freezing in the upper 300 m of the cloud (Fig. 5d). Similar to the open ocean case this radiative cooling and higher $N_{ice}$ in the perturbed simulations

favor increased depositional growth. However, the depositional growth rate in *ice_1000CCN* is only one third of the growth rate simulated in *ocean_1000CCN* (Fig. 6d).

    As evident from Fig. 7a, over the open ocean the elevated LWP decreases after reaching its maximum and returns to the LWP range of *ocean_control*. Independent of the strength of the CCN perturbation, LWP in all simulations relaxes back to the unperturbed state over the open ocean. On the contrary, over sea ice any CCN perturbation >200 $\mathrm{cm}^{-3}$ perturbs LWP and

IWP outside their simulated background conditions beyond 20 h simulation time. We relate this different aerosol response of the stratocumulus cloud over the ocean and stratus cloud over sea ice mainly to differences in cloud dynamics. Over the open ocean, the cloud response to CCN perturbations is shifted from the liquid to the ice phase, where the strong and rapid increase

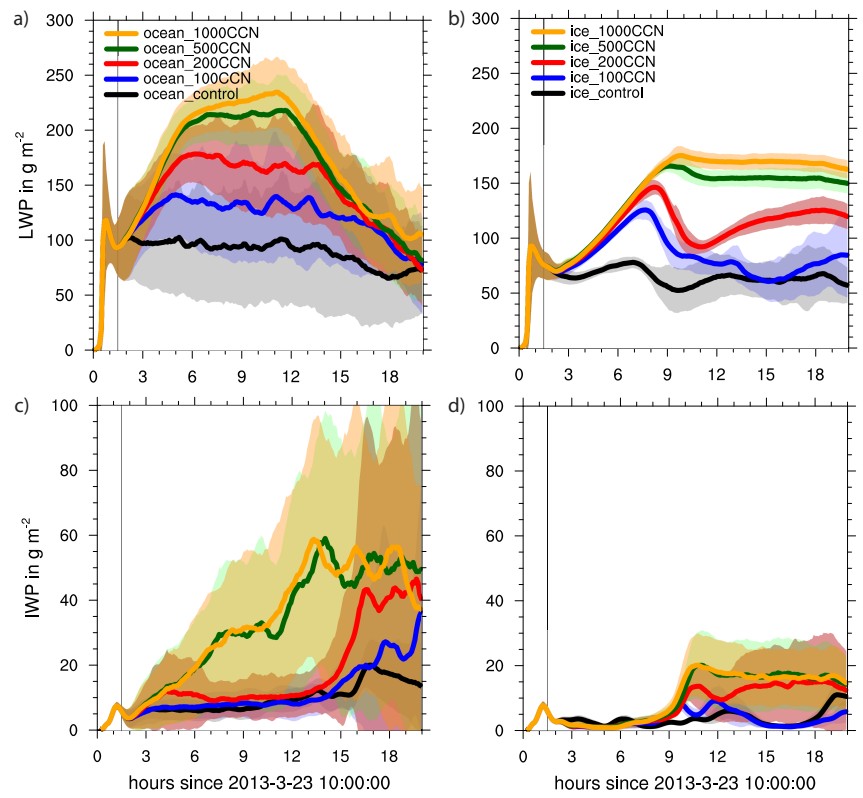

**Figure 7.** Domain averaged a,b) LWP and c,d) IWP over the open ocean (left) and sea ice (right) in the *control* and all CCN sensitivity simulations. The solid lines depict the means, the shadings the standard deviations. The vertical black lines indicate the CCN perturbation injections.

in ice mass reduces the liquid-phase response (Fig. 7c and Fig. S5a). Due to the increase in cloud ice and snow, increased surface precipitation after 12 h simulation time in the perturbed simulations additionally adds to the attenuated CCN response over the open ocean (Fig S1a).

Fig. 8 visualizes the spatio-temporal evolution of LWP within the domain over the open ocean. In the first hours after the initiation of the perturbation the LWP throughout the domain and within the updraft towers is increased (Fig. 8b). However, towards the end of the simulation, the cloud organizes back to structures similar to those observed in the control simulation (Fig. 8a). This behavior is qualitatively similar to what has previously been observed in numerical aerosol-perturbed simulations of warm-phase shallow cumuli (Jiang et al., 2006; Seifert et al., 2015). There, evaporative processes caused the limited sensitivity of the cloud field to aerosol perturbations. In our study, the main mechanism controlling the liquid-phase response of the stratocumulus cloud is the increased ice and precipitation formation.

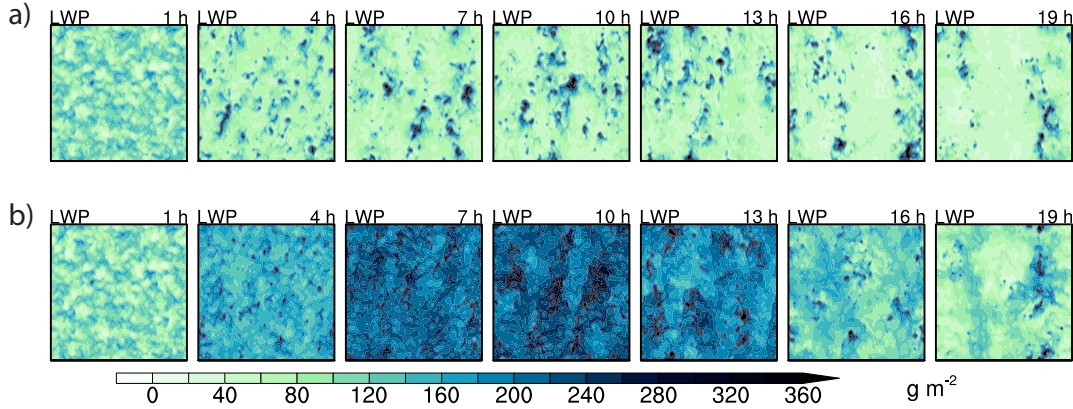

**Figure 8.** LWP for a) *ocean_control* and the b) open ocean *1000CCN* simulation.

## 5.2 Response to INP perturbations

Similar to the CCN perturbation simulations, we applied two INP perturbations of 3 and 10 INP $L^{-1}$ after 1.5 h simulation time. INP concentrations of over 10 $L^{-1}$ are not uncommon in Arctic spring conditions, representing Arctic haze (Rogers et al., 2001). In both dynamic regimes, IWP increases and LWP decreases with more available INPs (Fig. 9). As a result, the amount of precipitating ice and snow is increased in the perturbed simulations, while the amount of rain is decreased (not shown), similarly to the simulations perturbed by CCN. Total surface precipitation is increased within 2 h following the INP injections to 1.93 $\mathrm{mm\,d^{-1}}$ over the open ocean and 2.20 $\mathrm{mm\,d^{-1}}$ over sea ice in the *10INP* simulations, but thereafter not substantially impacted (Fig S1c,d).

The relative impact of INP perturbations is considerably larger than compared to CCN perturbations. A perturbation of 3 INP $L^{-1}$(i.e. an increase equal to the background concentration) doubles the peak IWP over the ocean from 5 to 10 $\mathrm{g\,m^{-2}}$, and decreases LWP by 12% from 100 to 88 $\mathrm{g\,m^{-2}}$ (Fig. 9a,c) one hour after INP injection. An equivalent change of CCN in *ocean_100CCN* increases LWP by merely 13% and does not (yet) increase IWP (section 5.1). Over sea ice, IWP increases initially by almost 300% from 3 to 12 $\mathrm{g\,m^{-2}}$ and LWP decreases also by 12% from 66 to 58 $\mathrm{g\,m^{-2}}$ for a perturbation of 3 INP $L^{-1}$ as compared to *ice_control* (Fig. 9d).

Considering the full simulation period, the mean IWP increase through INP perturbations remains below the response of the ice phase to CCN perturbations of 500 $\mathrm{cm^{-3}}$ or higher (Fig. 7c,d and Table 3). After investigating this increase in the ice phase in clouds with perturbed INPs, we conclude that in the *3INP* and *10INP* simulations ice crystal growth at the expense of liquid water through the WBF process (as seen in the increase in IWP accompanied by a LWP decrease) as well as changes in $N_{ice}$ (Table 3) through immersion freezing on INPs dominate the total IWP increase. The higher $N_{ice}$ follows the Twomey effect in the sense that $R_{ice}$ is smaller (Table 3), but IWP is still increased (Kärcher and Lohmann, 2003). This is insufficient

to exceed the IWP increase in clouds perturbed by CCN, where growth by deposition in the colder and destabilized cloud layer dominates any changes in $N_{ice}$.

Also, even though the relative impact of INP perturbations is large, in neither regime does a perturbation of 10 INP L$^{-1}$ glaciate the cloud. This finding is consistent with other studies investigating cloud glaciation under INP perturbations (e.g. Morrison et al., 2008; Solomon et al., 2018), but in contrast to Young et al. (2017), who simulate cloud glaciation using different (but more simplified) ice nucleation parameterizations for the same case. Considering $N_{drop} \gg N_{ice}$ throughout the simulation, a complete glaciation of the cloud seems surprising with an INP perturbation of only 10 L$^{-1}$.


The stratus cloud over sea ice is initially very susceptible to INP perturbations, which induce an initial peak in IWP and surface precipitation before the cloud returns to the unperturbed state. However, the more dynamic cloud structures over the open ocean are able to maintain an elevated IWP by 300% throughout the simulation.

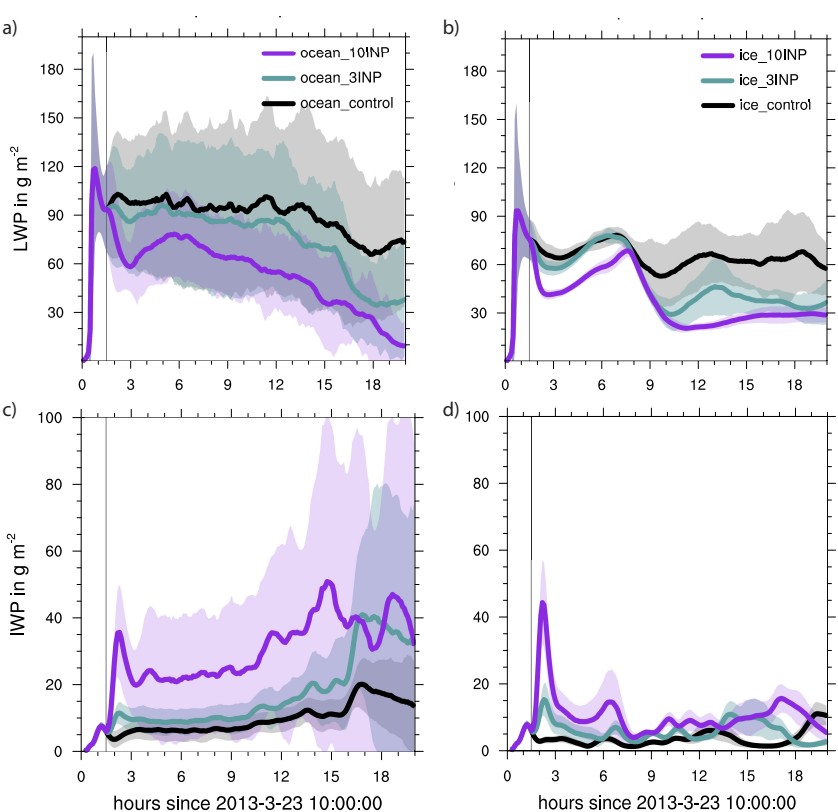

**Figure 9.** Domain averaged a,b) LWP and c,d) IWP over the open ocean (left) and sea ice (right) in *control* and all INP sensitivity simulations. The solid lines depict the means, the shadings the standard deviations. The vertical black lines indicate the CCN perturbation injections.

## 5.3 Sensitivity to different temperature regimes

To address the robustness of our conclusions to different temperature ranges, we performed the *control*, the *1000CCN*, and the *10INP* simulations over sea ice and open ocean in 2 K warmer and colder conditions. The relative humidity was kept constant.

The environmental conditions mainly determine the partitioning of moisture between the liquid and the ice phase (Fig. 10). Focusing on the open ocean case first, the response to CCN perturbations is intensified in the cloud liquid phase under warmer conditions, as LWP increases compared to *ocean_1000CCN* and IWP decreases. This is of course related to the fact that at

warmer temperatures less INPs nucleate, which decreases $N_{ice}$ (Fig. 10c and Table S2). In contrast, at colder temperatures more INPs nucleate, IWP increases earlier on as in *ocean_1000CCN* and LWP is considerably reduced (Fig. 10a,c and Table S2). However, even under warmer conditions LWP in *ocean_1000CCN+2K* relaxes to its unperturbed state and returns to the range of *ocean_control* at the end of our simulated time period (Fig. 10a). Hence, our conclusion concerning the buffered aerosol response in the liquid phase over the open ocean remains valid for warmer environmental conditions.

Over sea ice the aerosol response of LWP is also sensitive to the environmental conditions. Under warmer conditions, the cloud shows a similar behavior to the open ocean case. LWP in the *ice_1000CCN+2K* shows a similar increase to the *ocean_1000CCN* case and relaxes to the unperturbed conditions after 18 h. The temporal evolution of the LWP (Fig. S6) indicates small convective cells between 4-16 h in the *ice_1000CCN+2K* simulation in contrast to *ice_1000CCN*. As ice processes play a minor role in the *ice_1000CCN+2K* simulation, a strong precipitation event around 13-14 h likely causes the

LWP to relax back to the unperturbed state (Fig. S1f).

For INP perturbations, the temperature change initiates increased freezing and a higher IWP for the colder simulations and vice versa for the warmer simulations (Fig. S7). Determined by the nature of the DeMott et al. (2015) immersion freezing parameterization that is based on observations, more (less) INP nucleate at colder (warmer) temperatures.

## 6 Discussion

To summarize the cloud micro- and macrophysical responses to both, INP and CCN perturbations, we calculated the mean cloud properties in Table 3 (and Table S1 for all CCN and INP perturbation simulations not listed in Table 3). Additionally, a schematic of our findings is shown in Fig. 11. The first panels in each row conclude our results from section 4, indicating the existence of two different cloud regimes, a stratocumulus regime over open ocean and a homogeneous stratus regime over sea ice. These distinct regimes mainly result from differences in updraft speed, leading to different efficiencies in vertical moisture

transport, subsequent cloud droplet growth, precipitation and ice formation. Our results agree with previous findings obtained from satellites and measurement campaigns as well as the ACCACIA observations and modeling results. As has been observed by Young et al. (2016) and simulated by Young et al. (2017), we also simulate a MPC over the ocean with a higher cloud top, larger droplets, increased LWP and IWP and increased precipitation rates. The development of cumuli over the ocean as a response to increased surface fluxes additionally supports findings by Young et al. (2018).

As in Schweiger et al. (2008) and in agreement with previous ACCACIA studies our results indicate a higher cloud base over the open ocean and geometrically thicker clouds than over sea ice (supporting findings by Palm et al., 2010). Similarly to

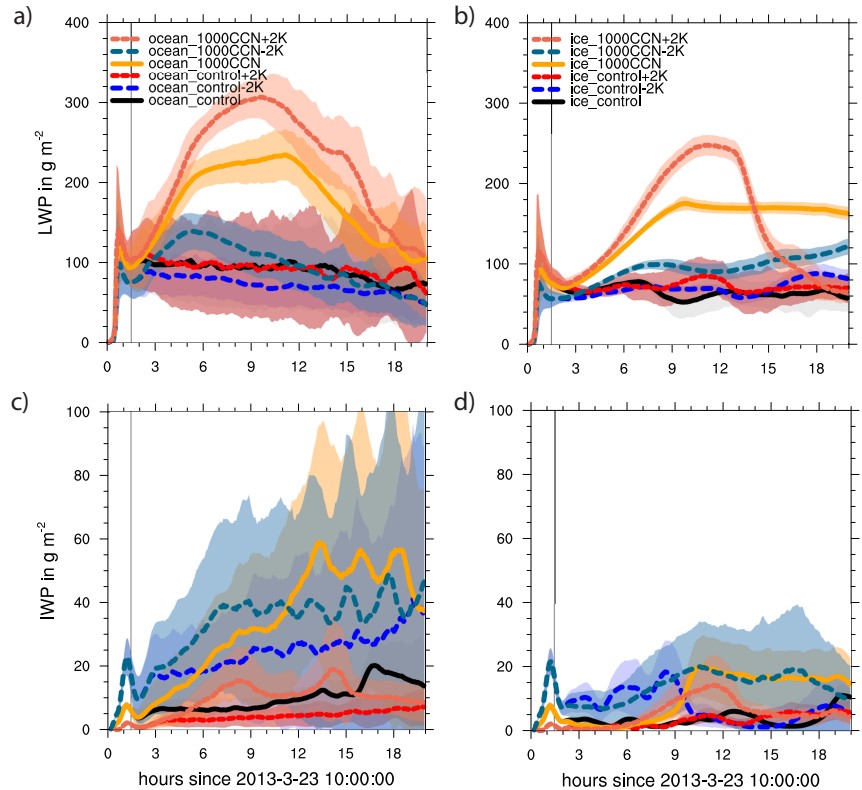

**Figure 10.** Domain averaged a,b) LWP and c,d) IWP over the open ocean (left) and sea ice (right) in *control* and the respective *1000CCN* simulations in their regular state and 2 K warmer and colder conditions. The solid lines depict the means, the shadings the standard deviations. The vertical black lines indicate the CCN perturbation injections.

Sotiropoulou et al. (2016) we also note structural differences over both surfaces with a stratocumulus cloud regime over the ocean versus a stratus cloud over sea ice. However, while Sotiropoulou et al. (2016) relate changes in cloud properties mainly to changes in atmospheric stability over the open ocean and sea ice, our case studies are initialized with the same atmospheric

stability profile, hence we suggest that the differences in surface latent and sensible heat fluxes may play a stronger role than previously suggested. In terms of radiative effects, the cloud over the open ocean and sea ice have different impacts on the net surface radiative balance. Note that the prescribed surface emissivity for ocean and sea ice is unchanged in both simulations. However, due to the 3 K warmer ocean, the LW surface emission is slightly increased over the open ocean and was quantified as $2.4 \pm 1.1 \ \mathrm{W \, m^{-2}}$ (spatiotemporal average over the first cloud-free hour). Additionally, we find cloud base height to be the

dominating factor determining the net surface LW radiative balance for clouds sufficiently optically thick in the LW spectrum (LW and SW radiation fluxes are defined to be positive downwards throughout our study). As the cloud over sea ice has a lower cloud base, the cloud re-emits LW radiation at warmer temperatures, which reduces the net surface LW cooling (Table 3). The net surface SW radiation is directly coupled to cloud optical depth and by around $4 \ \mathrm{W \, m^{-2}}$ lower over the ocean, where the

optically thicker cloud reflects incoming solar radiation more efficiently. Hence, we can extrapolate that during months with sufficient incoming solar radiation, clouds over the ocean might have a net zero to a cooling effect as compared to clouds over sea ice (as also found by Gilgen et al., 2018).

In a next step, we applied aerosol perturbations to the two contrasting cloud regimes. As our model setup allows for a prognostic treatment of aerosol-cloud interactions, we are able to quantify the cloud response to spatio-temporally resolved aerosol perturbations, which is a novel aspect as compared to previous ACCACIA modeling studies (Young et al., 2017, 2018). Both studies Young et al. (2017) and Young et al. (2018) used a prescribed $N_{drop}$ concentration and parameterized $N_{ice}$ concentrations (not considering interactive INPs) in their model setup, which have been adjusted in sensitivity simulations by Young et al. (2018). In their study, the authors found smaller droplets in a simulation with increased $N_{drop}$, but found little effect on LWP or IWP. In contrast, we see a strong initial sensitivity of Arctic MPCs to CCN perturbations. Over ocean and sea ice, the LWP is already substantially increased with a perturbation of 200 and 100 CCN $\text{cm}^{-3}$, respectively. With increasing CCN perturbations, $N_{drop}$ ($R_{drop}$) increases (decreases), accompanied by an increase in LWP (in agreement with Morrison et al., 2008; Possner et al., 2017; Solomon et al., 2018; Stevens et al., 2018). As a result of the larger LWP, LW cooling increases in the perturbed simulations throughout the cloud and the cloud deepens, such that more ice crystals nucleate through increased immersion freezing. Additionally, ice crystals grow by enhanced deposition rates in the perturbed simulations. This increased IWP in simulations solely perturbed by CCN was noted before by Possner et al. (2017) as well as Solomon et al. (2018). As a result of higher IWP and LWP, the cloud becomes optically thicker, reflects more SW radiation and reduces the LW emission from Earth's surface (Table 3). This has a net cooling effect over daytime and during months with incoming solar radiation, but we expect the warming effect to dominate during polar winter.

Changes in the LW radiative properties are overall only moderate between the *control* and *1000CCN* simulations, ranging from 6-13% over sea ice and ocean, respectively. Most likely the change in cloud structure between the two regimes determines the smaller response in net surface LW radiation to CCN perturbations over sea ice than over the open ocean. The temporary transition from a stratocumulus to a stratus cloud over the ocean for a perturbation of 1000 CCN $\text{cm}^{-3}$ (Fig. 8) increases the cloud re-emittance throughout the domain. On the contrary, the additional thickness of the stratus cloud over sea ice has a smaller effect, as the cloud structure is not considerably changed. Interestingly, the change in cloud base as simulated between *ocean_control* and *ice_control* has a stronger LW radiative effect on the Earth's surface (4.3 $\text{W}\,\text{m}^{-2}$) than CCN perturbations of 1000 $\text{cm}^{-3}$ (3.4 $\text{W}\,\text{m}^{-2}$ over the ocean and 1.3 $\text{W}\,\text{m}^{-2}$ over sea ice). On the other hand, the increased optical thickness of the perturbed clouds increases the reflectivity of the cloud and reduces the net surface SW radiation by 33-45% over sea ice and ocean, respectively. This effect is larger than changes in net surface LW radiation, but is only important during daytime and spring to fall.

There is a strong regime-dependence of the MPC response to CCN perturbations, which is novel in the context of aerosol-cloud interactions. Over sea ice, the cloud evolution remains substantially changed throughout the simulation period for any CCN perturbation >200 CCN $\text{cm}^{-3}$. Over the open ocean, ice formation and growth as well as an increase in precipitation buffer the LWP response and lead to a relaxation of the LWP to its unperturbed state after 18 h simulation time. The cloud microphysical

properties such as $N_{drop}$ and $R_{drop}$ remain perturbed. This results in a sustained Twomey brightening of the cloud even 18

h following the CCN perturbation. Combined with a lowering of the cloud base, the outgoing surface LW is reduced by 2.5 W m$^{-2}$ and the incoming SW radiation by 3.5 W m$^{-2}$ during the last two simulated hours in *ocean_1000CCN*. The sustained net cooling is considerably smaller as compared to cooling rates simulated during the whole period (Table 3), but indicates a remaining perturbation of the cloud radiative properties. Additional observations such as the ACCACIA campaign, but in polluted environments, could help to constrain such regime-dependent aerosol-cloud interactions. Also further model studies

including prognostic aerosols could expand our findings to a wider range of meteorological conditions (which we touched upon with our temperature change sensitivity tests).

The initial relative impact of increasing INP concentrations is larger as compared to CCN concentrations. With more potential INPs, more particles are available for ice crystal formation by immersion freezing, which increases $N_{ice}$ and IWP. The increase in IWP is accompanied by a decrease in LWP through the removal of liquid water by deposition via the WBF process.

This is consistent with previous studies investigating the effect of increasing INP or $N_{ice}$ on Arctic MPCs (Morrison et al., 2008; Ovchinnikov et al., 2014; Stevens et al., 2018; Young et al., 2018). The lower LWP in the simulations perturbed by INPs leads to an optically thinner cloud in the *10INP* simulations which increases net LW cooling at the Earth's surface, but has little effect on the net surface SW radiation (Table 3). This is an opposing effect to CCN perturbations, which generally have a moderate LW warming and a strong SW cooling effect on the underlying surface. Interestingly, the IWP increase for a pertur-

bation of 10 INP L$^{-1}$ is smaller than the IWP increase in the *1000CCN* simulation over the open ocean (Table 3). We relate this difference to more efficient ice crystal growth by deposition in *1000CCN* than in *10INP*, supported by higher deposition rates (not shown) in experiments perturbed by CCN (Table 3). The stratus cloud over sea ice initially shows a stronger response to INP perturbations than the stratocumulus cloud over open ocean. This different sensitivity to INP changes between surfaces is consistent with findings from Morrison et al. (2008). Similarly, Jiang et al. (2000) found Arctic stratus over sea ice to be

specifically vulnerable to INP perturbations. Note that with time, the IWP increase (LWP decrease) is more pronounced over the ocean, which can be related to stronger updrafts and cooling as well as the continuous cloud deepening over open ocean.

**Table 3.** Averaged cloud properties ±1 standard deviation throughout the simulated time period following the aerosol injection (hour 2-20) for the unperturbed and perturbed simulations. Note that for net surface SW radiation we only averaged over daytime (8.5 h in total).

| | ocean_control | ocean_1000CCN | ocean_10INP | ice_control | ice_1000CCN | ice_10INP |
|---|---|---|---|---|---|---|
| Cloud mean $N_{drop}$(cm$^{-3}$) | 48.1±15.4 | 306.6±68.5 | 52.0± 15.4 | 40.1±18.4 | 230.5±53.6 | 39.1±15.4 |
| Cloud mean $R_{drop}$(μm) | 6.5±1.7 | 4.1±1.0 | 6.1±1.6 | 5.8±1.8 | 4.4±1.0 | 6.4±2.0 |
| LWP (g m$^{-2}$) | 89.8±50.9 | 176.5±57.3 | 52.8±31.4 | 64.9± 17.7 | 147.6± 34.2 | 37.0±14.8 |
| Cloud mean $N_{ice}$(L$^{-1}$) | 0.27±0.20 | 0.44±0.34 | 0.84±0.75 | 0.08±0.05 | 0.17±0.10 | 0.27±0.19 |
| Cloud mean $R_{ice}$ (μm) | 15.5±2.0 | 17.5±3.2 | 14.4±2.6 | 18.0±2.9 | 18.4±3.4 | 15.9±2.6 |
| IWP (g m$^{-2}$) | 10.1± 7.1 | 35.2± 34.8 | 31.6±34.7 | 3.6± 2.8 | 11.0±9.9 | 10.8±8.0 |
| Cloud optical depth | 9.5±4.9 | 28.9±8.8 | 5.8±3.3 | 7.6±2.2 | 23.0±5.1 | 4.0±1.7 |
| Net surface LW (W m$^{-2}$) | -25.1±3.9 | -21.7±4.0 | -28.6±11.5 | -20.8±4.5 | -19.5±4.4 | -26.4±6.5 |
| Net surface SW (W m$^{-2}$) | 19.1±14.3 | 10.5±9.7 | 20.3±14.7 | 24.1±18.6 | 16.1±14.2 | 26.4±19.4 |

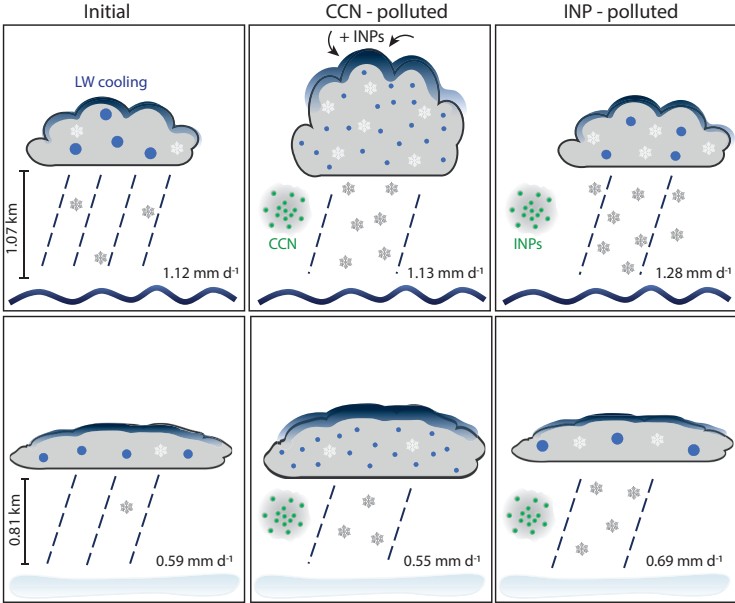

**Figure 11.** Conceptual overview of the cloud response to increased CCN and INP concentrations. The first row illustrates the open ocean stratocumulus regime, the lower row the stratus over sea ice.

## 7 Conclusions

The analysis of MPCs within a changing Arctic environment has been the subject of a number of recent studies (Browse et al., 2014; Christensen et al., 2014; Young et al., 2016; Possner et al., 2017; Gilgen et al., 2018). Here, we addressed the cloud properties of MPCs in two differing regimes (i.e. sea ice and open ocean) in a series of high resolution LES. The robustness of the response to an aerosol perturbation was evaluated by applying our perturbation scenarios in warmer/colder environmental conditions.

Our key findings are summarized as follows:

1. The surface properties have a considerable impact on MPC properties. Our simulations support previous results obtained for the ACCACIA campaign (Young et al., 2016, 2017, 2018): over the open ocean, strong turbulent surface fluxes increase the updraft velocities, which in turn favor the development of cumuli towers feeding moisture into the stratus layer. This increased vertical moisture flux leads to an increase in the cloud LWP and IWP, larger cloud droplets and ice crystals and a higher cloud base and cloud top. Over sea ice, surface fluxes and in turn updraft velocities are low, which confines the cloud to a homogeneous stratus cloud. As the boundary layer is generally less moist, cloud droplet and ice crystal formation and growth are limited as compared to the cloud over open ocean.

2. Aerosol perturbations providing potential CCN substantially impact the cloud LWP and IWP immediately after the perturbation injection. The MPC over the ocean responds with an increase in $N_{drop}$ and LWP. Through increased LW cooling throughout the cloud, new ice crystal formation by immersion freezing and subsequent growth by vapor deposition, IWP increases. Over sea ice, CCN activation is less efficient and the maximum response is delayed and weakened.

3. The relative initial response of the cloud to INP perturbations is larger than to CCN perturbations. The response is relatively straightforward and agrees with previous results. INP perturbations immediately increase the IWP and decrease the LWP in both cloud regimes. In our simulations, none of the applied INP perturbations (3 and 10 $L^{-1}$) is sufficient to cause complete cloud glaciation.

4. The cloud response to aerosol perturbations is highly regime-dependent. Over the open ocean, LWP perturbations are efficiently buffered after 18 h simulation time. Increased ice and precipitation formation relax the LWP back to its unperturbed range. Over sea ice the cloud evolution remains substantially perturbed with CCN perturbations ranging from 200 to 1000 CCN $cm^{-3}$. For INP perturbations, an intense ice formation and precipitation peak is triggered with no further subsequent change in cloud properties. Over the open ocean, LWP and IWP remain perturbed throughout the simulation for an INP perturbation of 10 $L^{-1}$.

Extrapolating our findings to a future ice-free Arctic, increased ship traffic, and higher levels of pollution at the high latitudes, we find that changed surface conditions are likely to highly affect MPC dynamics, properties and hence the radiative budget of the surface. The effect of pollution will be most effective in stratiform clouds over sea ice, where INP perturbations on the order of 10 $L^{-1}$ lead to a strong cloud thinning and thus a change of the radiative balance on the order of a 4 $W\,m^{-2}$ cooling at

the surface. Similarly, CCN perturbations may also cool the underlying surface through increased reflection of incoming SW
radiation, but might have a warming effect in the absence of solar radiation. Considering that ship exhaust plumes may consist
of both, CCN and INPs (Hobbs et al., 2000; Thomson et al., 2018), the combined aerosol effect on Arctic MPCs may offset
Arctic warming during the summer months, but we are doubtful it completely counteracts Arctic warming during the full year
as also suggested by Christensen et al. (2014) and Possner et al. (2017).

Nevertheless, we note that our study has come caveats. We used the open ocean initial dropsonde profile to initialize both
our cases (open ocean and sea ice), which is in contrast to Young et al. (2017). Over vast sea ice covered surfaces the boundary
layer profile might highly differ from the boundary layer over open ocean (Young et al., 2016) and thus the clouds may evolve
differently. However, we wanted to narrow possible differences over open ocean and sea ice case down to surface fluxes, which
become important over freshly melted sea ice or polynyas (Gultepe et al., 2003). In addition, due to runtime limitations it was
not possible to simulate these high-resolution simulations for a longer time period. Thus, we unfortunately cannot draw any
conclusions concerning cloud stability and persistence beyond 20 h.

*Data availability.* The COSMO-LES model simulations are stored at the Swiss National Supercomputing Center and can be made accessible
upon request.

*Author contributions.* GKE conducted the simulations, analyzed the results, and was the main author of the paper. AP and UL contributed
to the design of the study and the analysis of the results. All authors contributed to the writing of the study.

*Competing interests.* The authors declare that they have no conflict of interest.

*Acknowledgements.* The research leading to these results has received funding from the European Union's Seventh Framework Program
(FP7/2007-2013) project BACCHUS under grant agreement No 603445. AP is receiving support from MOPGA-GRI (57429624) which
is funded by the BMBF and implemented by the DAAD. All simulations were performed with the Consortium for Small-scale Modeling
(COSMO) model adapted for large eddy simulations. The simulations were performed and are stored at the Swiss National Supercomputing
Center (CSCS). The ACCACIA observations were obtained from the NCAS British Atmospheric Data Center (http://catalogue.ceda.ac.uk/uuid/88f95b1d5
We thank Gillian Young for providing the Young et al. (2017) LES model runs. Finally, we thank the three anonymous reviewers who made
very useful comments that strengthened the manuscript.

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
