# Peer review of "Response of Arctic mixed-phase clouds to aerosol perturbations under different surface forcings"

_Atmospheric Chemistry and Physics, 2018_

## Referee Comment (RC1) · Anonymous Referee #1 · 3 Dec 2018

The study titled "**Relaxation times of Arctic mixed-phase clouds to short-term aerosol perturbations under different surface forcings**" by Eirund et al. illustrates how mixed-phase clouds, modelled using large-eddy simulations, respond microphysically to short bursts of high aerosol number concentrations, such as those which would be experienced in the vicinity of shipping emissions. By simulating two cloud scenarios – one over sea ice, the other over ocean – the authors show how the surface conditions, moderated by chosen sensible and latent heat fluxes, can affect how the clouds respond to the influx of high aerosol particle number concentrations.

The study builds upon previous work using measurements from the Aerosol-Cloud Coupling And Climate Interactions in the Arctic (ACCACIA) campaign, and tests these observations with a more complex model representation of aerosol-cloud interactions than has been done before; therefore, the results are an important addition to the scientific literature. However, before publication, I have a few concerns which I feel should be addressed. The study has some potential implications for the stability of Arctic clouds in the face of increased shipping emissions across the region; however, these are not suitably discussed at present. Furthermore, the authors come close to repeating some conclusions from Young et al., 2016a and 2017, and should distinguish the novelty of their results more so from these published works.

**General comments:**

1. The paper does not suitably cite previous work on this case study from the ACCACIA campaign. The work is novel in its prognostic representation of both CCN and INP; however, similar studies have been already conducted which compare cloud microphysics over sea ice and over ocean. Please ensure all previous literature is cited more appropriately. For example, the differences between the boundary layer structure over sea ice and over ocean has been discussed as an observational study by Young et al., 2016a, and large-eddy simulations of this case study have been presented by Young et al., 2017. Observational conclusions should not be repeated here as conclusions of this study unless these earlier works are cited appropriately.
2. The same boundary layer properties are used to compare how clouds form over the ocean or sea ice from the same state. Whilst this is an interesting perspective, why did the authors not use a boundary layer profile measured over the sea ice for these simulations? Do the authors expect the resulting cloud to compare well with observations when the initial profiles used are not the same as that measured? The boundary layer over sea ice is different to that over the ocean (as presented by Young et al., 2016a; 2017); therefore, can the authors comment on why they used the oceanic profile for the sea ice simulations?
3. Readability and clarity could be improved – for example, it is often not clear whether the model simulation results or measurements are being discussed.
4. The Discussion section could be significantly improved – it currently focuses on validating findings against previous studies; however, there is an opportunity to compare with previous ACCACIA studies which is currently not being capitalised upon. Specifically, there is an opportunity to conduct comparisons with the LES findings of Young et al., 2017 – your results are similar, and therefore there is scope to make some preliminary statements about the ability of two different models to reproduce these observations.
5. The INP perturbation experiments are lacking analysis and discussion. Please add to this section of the study or remove it.
6. The authors have the opportunity to make some preliminary statements about the stability and microphysical response of Arctic MPCs in the face of pollution transport/shipping emissions (as suggested in the Introduction); however, little discussion of this is included. Please comment on the potential real-world implications of this modelling study.

**Specific comments:**

**Abstract:**
**Page 1:**
**Line 4:** was ACCACIA conducted in the central Arctic? I would've taken this to be >80N?

**Line 5:** define COSMO

**Lines 6–11:** these findings read very similarly to those presented by Young et al., 2016a; 2017. Are they conclusions from your modelling work? It currently reads like conclusions from the measurements used to initialise the model – measurements that have already been published. Additionally, LES studies of this case study have already been published. Please distinguish your conclusions more so from these studies to highlight its novelty.

**Line 12:** "*two dynamically different regimes*" – this is quite vague, can you expand on this?

**Line 16:** "*the maximum response*" – response in what?

**Line 18:** Could you specify more about your aerosol perturbations here? For example, their duration / altitude?

**Line 20:** Can you say more here about how the aerosols are transported out of the boundary layer?

**1. Introduction:**
**Page 2:**
**Lines 13-15:** Please rephrase – sentence meaning unclear

**Line 16:** "*amount*" is vague – "*fraction*"?

**Line 23:** Please provide references for the sentence ending "*potential implications for cloud dynamics*".

**Line 23:** Please define ECMWF.

**Line 29:** please rephrase

**Line 34:** Can the Arctic still be called pristine? It is cleaner than the mid-latitudes and is very clean in the summer, but the Arctic haze strongly increases aerosol mass concentrations in the Arctic atmosphere during the spring, where the data used to initialise these model simulations was collected. Please consider rephrasing this statement.

**2. Model description and setup**
**Page 3:**
**Line 27:** Dropsondes were discussed in detail in Young et al., 2016a – please cite here for reference.

**Lines 29-31:** Spatial domain size and resolution are specified, please include similar information regarding run length and temporal resolution.

**Page 4:**
**1ˢᵗ paragraph:** Solomon et al., 2018 (ACP) use the DeMott et al, 2010 (PNAS) parametrization, not the DeMott et al, 2015 (ACP) parametrization. They use a prognostic derivation of the temperature-dependent fit from Fig. 2 of DeMott et al., 2010, removing the dependence on aerosol number concentration from the standard version of the parametrization. Please can the authors clarify which ice nucleating particle parametrization they used, and note whether they included the aerosol number concentration dependence or used the version (described in Solomon et al., 2015, ACP) which is dependent only on temperature? DeMott et al., 2015 (ACP) is designed for mineral dusts; therefore, if this relationship was used, can the authors comment on the validity of doing so in the Arctic where there is some dust (Young et al., 2016b, ACP) externally mixed with other aerosol species?

**Lines 7-8:** Are these median diameters that are quoted? Please clarify.

**Line 9:** "*low altitudes*" – please specify. Did the authors exclude in-cloud measurements with the PCASP (standard practice)? If so, how?

**2nd paragraph:**
Please say more about how the aerosol observations are used as input to the model. The PCASP only measures >0.13 micron, do you have any other aerosol measurements for the smaller size mode? When fitting these two modes to the observations, what geometric standard deviation was used for each? Was a lognormal distribution assumed? Did you fit to data collected over the entire ocean segment of the ACCACIA flight in question, and did you use different inputs for the sea ice simulations?

**Lines 10-11:** How did you arrive at this INP concentration? In Young et al., 2016a, the parametrizations listed are temperature dependent and were evaluated at the coldest temperature measured over the sea ice or ocean. Is the same technique used here? 3.3 $L^{-1}$ is higher than presented in Young et al., 2016a for PCASP data over both the sea ice and ocean (their Table 3), and seems particularly so given the sensitivity to ice particle number concentration presented by Young et al., 2017. Please provide more information about where this number concentration has come from.

**Line 14-15:** How did you arrive at these values? They are within the range measured (Young et al., 2016a), yet they are very specific choices which should be justified.

**2.1 Model perturbation experiments**
**Line 22:** are all model analyses taken after 1.5h? Is this taken to be the spin-up period of the model? From Fig. 6, it looks like the spin up may be until 2h? Can the authors discuss whether other diagnostics (such as TKE or W) were used to define the spin up? Additionally, how was the perturbation time chosen? Was this taken to be immediately after spin up (inferred from Fig. 6)?

**Lines 22-23:** Why is the pollution mode smaller? I agree it should be, but more could be done to justify this choice. Why was 0.19 micron specifically chosen? Again, is this the median diameter of the mode?

**Line 26:** This reads like you have both doubled the background and increased by a factor of 3 (i.e. ~16.6 $L^{-1}$) – please clarify

**Line 28:** Please provide references for this statement.

**Line 21:** How long is a single time step? (see previous comment on temporal resolution)

**3. Evaluation of background state**
**Page 5:**
**1st paragraph:** Here the authors seem to jump between the simulations and the initial conditions and discuss these interchangeably. Please describe the initial conditions (dropsonde measurements) first, then the controls to avoid confusion.

**Line 9:** *ocean_control* is used interchangeably to refer to the observations and the control model simulations. This label should refer to model results only. Please use a different label (e.g. *observations*) to describe the dropsonde measurements. Also, it lists the mean $N_{ice}$ is 0.17 $L^{-1}$ in Table 2?

**Lines 11-13:** Did you try changing the background concentration of CCN to improve the cloud microphysical agreement with observations? 3.39/3.99 $cm^{-3}$ is significantly less than observed – are these comparisons with the observations robust? There are significant differences in cloud base/top height and cloud properties, the study would therefore benefit from some discussion on why this is the case and how these differences may affect the real-world implications.

**Line 14:** Why do those aerosols in the inversion layer not activate? Is it sub-saturated? Is there too much competition for water vapour?

**Line 15:** Please provide a comment on the realism of this finding (poor re-entrainment of aerosols due to low turbulence).

**Lines 15-17:** Please clarify. Are you talking about the dry run where you have $N_{drop}$ formation? How is this possible if the boundary layer is kept below water saturation?

**Page 6:**
**Line 1:** Do you show this anywhere? (Mixing of CCN from above)? Or is it inferred from Fig. 2? Perhaps some w' tendencies could be shown to illustrate upwards/downwards motion (if these diagnostics are available)?

**4. Surface flux impact on cloud dynamics**
**Page 7:**
**Lines 1-2:** Young et al., 2018 discuss similar simulated effects from oceanic surface fluxes using these observations for initialisation – please consider cite/comparing here.

**5.1 Response to CCN perturbations**
**Page 9:**
**Line 6:** "*is sufficient to significantly perturb*" – please elaborate. By how much? Do smaller perturbations not change the cloud physics as much? There is an opportunity to discuss real-world consequences here.
**Line 15:** For reference to Fig. 6C to be valid, need to mention IWP here too.

**Page 10:**
**Line 7:** Young et al., 2018 showed that these detraining layers of moisture can be reduced by implementing strong large-scale subsidence. Cloud top increases with time in your ocean simulations due to the heat and moisture fluxes from below – have you looked at the effect of increasing your imposed subsidence to reduce cloud deepening?
**Lines 10-11:** Why? Opportunity to discuss.
**Line 14:** "*most perturbed simulation*" – please quote run label for clarity

**Page 11:**
**Line 4:** "*increase in the ice phase*" – please be more specific, do you mean number concentration? Mass concentration?
**Lines 5-6:** Even though the largest perturbation simulation LWP relaxes back to similar trends as the control simulation, it's magnitude is still approximately 2-3× that of the control. Given the low LWPs simulated, this small difference could have an effect on the radiative properties of the clouds. Please discuss.

**Page 12:**
**Line 1:** Figure 2b does not show transport out of the boundary layer, it just shows non-zero number concentrations. To prove transport, could you show some tendencies (perhaps some relationship with w')? Or perhaps a time series of aerosol particle number concentration profiles (like Figures S2-S4)?

**5.2 Response to INP perturbations**
**Page 13:**
**1st paragraph:** Could the authors show how the INP perturbations affect $N_{ice}$, in addition to LWP/IWP? Also, there is little analysis on this section's findings in comparison to the CCN perturbations, why is this? As the manuscript stands, this section reads like an afterthought.

**Lines 6-7:** Is this illustrated anywhere? If not, please include a figure (like Figure 7) in the supplementary as evidence.

**Side note for Discussion:** The authors show that the cloud does not glaciate (in agreement with other studies). These findings are in contrast to Young et al., 2017's ACCACIA LES results, who use a more simplified representation of cloud microphysics and aerosol-cloud interactions. Could this mixed-phase persistence be because the ice number concentrations are much lower than observed, and modelled in that case? There is an opportunity to compare with their findings, which the authors do not capitalise on. Also, there is a lack of analysis/discussion on the INP perturbation experiments – does the extra ice created by the INP injection precipitate out of the cloud as snow? If so, how does the INP injection affect precipitation rates?

**5.3 Invariance of results across temperature regimes**
**Page 14:**
**Lines 6-10:** Please improve clarity
**Line 12:** Again, is this the parametrization used? This is not the same as in Solomon et al., 2015.

**5.4 Consistent response independent of perturbation injection period**
**Page 15:**
**Lines 2-4:** This should be made clearer at the start – to me, it was not clear until now what the aerosol perturbations represented in model terms.

**Side note:** This section seems to be "in response" to some discussion, perhaps it should be relocated to a sub-section of Section 6?

**6. Discussion:**
**Page 16:**
**Line 5:** Reference to Young et al., 2016a – this study uses a range of aircraft measurements to show this, and model results here should be presented as successfully reproducing these conditions rather than new conclusions.
**Line 11:** As previous, did you try using a sea ice dropsonde profile? Like that presented in Young et al., 2017 for this case? These conclusions are very similar to the observational conclusions of Young et al., 2016a and modelling conclusions of Young et al., 2017. Please reference these studies here – there is a great opportunity to show how these results compare with the previous studies, especially since a more complex microphysical modelling representation is used here. There is novelty in these results; however, the distinction between conclusions from this study and those from previous ACCACIA work is not clear.
**Line 13:** "*possible pathway for cloud-aerosol interactions*" – what is meant by this statement? Not clear/vague.
**Line 15:** Define τ
**4th parapraph:** The authors refer to "*polluted*"/"*unpolluted*" – are you referring to the CCN perturbation experiments only? There is no reference to the INP perturbation experiments here, and there is a lack of analysis/discussion on these simulations.
**Lines 32-33:** It has been previously stated that the perturbation experiments relaxed back to their initial state but the authors have here clarified that there is some difference in magnitude (as per my previous point). This is confirmed in the values quoted in Table 3 between the controls and "post-polluted" rows. Please ensure analysis and discussion is consistent throughout the manuscript.

**Page 17:**
**Line 2:** Do you show aerosol particle transport out of the boundary layer? It is possible I have missed it, but non-zero values above don't necessarily show that aerosols are being transported vertically. Perhaps some microphysical tendencies (if you have the relevant diagnostics) could show the upward transport of aerosols? Or a time evolution of aerosol number concentration (like Figures S2-S4)? As it stands, this statement does not seem to be supported by any figure in the manuscript or supporting information.
**Lines 5-7:** There are more caveats to this study than listed here. For example, the fact that a sea ice boundary layer profile was not used for the sea ice simulations is a significant caveat that requires discussion. Why was this not used?

**7. Conclusions:**
**Page 19:**
**Line 2:** References for "*… the subject of a number of recent studies*"
**Lines 8-10:** Opportunity to link with Young et al., 2018 ACCACIA study (cumuli tower development – inter-model agreement)
**Lines 8-14:** Make stronger links to previous ACCACIA studies and make novelty of results more distinct from previously published conclusions.
**Line 18:** "*Over sea ice, cloud droplet growth is less efficient...*" – why? There has been little discussion of why microphysical processes occur differently over sea ice and ocean.
**Lines 25-26:** How? Please provide details.

**Line 28:** "*possible pathway for aerosol-cloud interactions*" – this statement has been used before and the meaning is not clear. Do the authors just mean that aerosol plumes may affect cloud structure in the Arctic? Please clarify.

**Technical corrections:**

**Page 1, line 19:** typo → "**properties**"
**Page 2, line 2:** "high **model** uncertainties"
**Page 2, lines 12**: "**mid-latitude**"
**Page 4, line 23:** "to be **at** slightly smaller **sizes** than"
**Page 4, line 25:** "successively" → "**in stages**"
**Page 5, line 8:** "according to" → "**agreeing with**"
**Page 5, line 15:** "In a dry run,…" – new paragraph?
**Page 7, line 7:** "allow the **cloud** droplets as well as the **ice** crystals"
**Page 7, line 15:** "LW cooling is increased up to… " → "**LW cooling increases up to**… " – the former reads like you are modifying the LW cooling, not a simulated effect.
**Page 7, lines 15-16:** "The more numerous ice crystals" → "**Higher concentration of** ice crystals"
**Page 8, Figure 4 caption:** typo → "**interquartile**"
**Page 9, line 11:** "upon seeding" → "**after seeing**"
**Page 10, line 8:** "The initial… on the cloud regime" – remove, vague and not required.
**Page 10, line 9:** "cloud droplet growth **is** limited"
**Page 19, line 19:** should this be a new paragraph?
**References:** Some references are incomplete or incorrect.

**Figures and Tables:**

**Table 1:** Please list columns as "Background CCN/INP".
**Table 2:** Total values taken over how long? The entire run? Excluding spin up? Please clarify.
- Caption: "*Note that the airplane did not sample the lower and upper levels*" – of what? The model domain? Please clarify/rephrase
- in-cloud criteria: both the liquid and ice mass thresholds? Or just one or the other? Please clarify, and define $q_c$ and $q_i$
- Can you comment on the very low cloud base height with comparison to the observations over the ocean? Or the cloud top height which is almost double the altitude of that observed over the sea ice?

**Please increase legend size on all figures.**

**Figure 1:**
- please choose different colours – hard to read
- improve readability – perhaps split into 4 panels? Over ocean/sea ice?

**Figure 2:**
- How do these profiles compare with observations?
- Would it be clearer to have: ice control (black), ice perturb (grey), ocean control (red), ocean perturb (orange)?

**Figure 3:**
- There is no increase in ice number with decreasing altitude like in the observations (Young et al., 2016a ACP), please comment on this. Similarly for the LWMR – these trends are in contrast to those observed, please comment on why.
- In caption, define LWMR

**Figure 5:**
- Why are the panels not shown to 0 m? Or at least to cloud base?

**Figure 6:**

- it may be beneficial to show Fig. S4 as additional sub-panels of Fig. 6 to show how the cloud structure evolves with time in the different scenarios

**Figure 7:**
- Just a side note, this figure does not print well (not clear which line is which). Consider changing colours used, or splitting into sea ice/ocean sub-panels?

**Figure 10:**
- This figure is particularly crowded and individual traces are hard to distinguish. Perhaps separate into further sub-panels? (e.g. ocean+1000CCN, ocean control, ice+1000CCN, ice control)?

**Figure 12:**
- Is precipitation always as rain? Again, do you refer solely to the CCN perturbation experiments for "polluted" case analysis. Please define what you mean by "polluted", and include some analysis on the INP perturbation experiments (or remove).

**List of referenced works:**

DeMott, P. J., Prenni, A. J., Liu, X., Kreidenweis, S. M., Petters, M. D., Twohy, C. H., Richardson, M. S., Eidhammer, T., and Rogers, D. C.: Predicting global atmospheric ice nuclei distributions and their impacts on climate, P. Natl. Acad. Sci. USA, 74, 2293–2314, https://doi.org/10.1073/pnas.0910818107, 2010.

DeMott, P. J., Prenni, A. J., McMeeking, G. R., Sullivan, R. C., Petters, M. D., Tobo, Y., Niemand, M., Möhler, O., Snider, J. R., Wang, Z., and Kreidenweis, S. M.: Integrating laboratory and field data to quantify the immersion freezing ice nucleation activity of mineral dust particles, Atmos. Chem. Phys., 15, 393–409, doi:10.5194/acp-15-393-2015, 2015.

Solomon, A., Feingold, G., and Shupe, M. D.: The role of ice nuclei recycling in the maintenance of cloud ice in Arctic mixed-phase stratocumulus, Atmos. Chem. Phys., 15, 10631–10643, https://doi.org/10.5194/acp-15-10631-2015, 2015.

Solomon, A., de Boer, G., Creamean, J. M., McComiskey, A., Shupe, M. D., Maahn, M., and Cox, C.: The relative impact of cloud condensation nuclei and ice nucleating particle concentrations on phase-partitioning in Arctic Mixed-Phase Stratocumulus Clouds, Atmos. Chem. Phys. Discuss., https://doi.org/10.5194/acp-2018-714, in review, 2018.

Young, G., Jones, H. M., Choularton, T. W., Crosier, J., Bower, K. N., Gallagher, M. W., Davies, R. S., Renfrew, I. A., Elvidge, A. D., Darbyshire, E., Marenco, F., Brown, P. R. A., Ricketts, H. M. A., Connolly, P. J., Lloyd, G., Williams, P. I., Allan, J. D., Taylor, J. W., Liu, D., and Flynn, M. J.: Observed microphysical changes in Arctic mixed-phase clouds when transitioning from sea ice to open ocean, Atmos. Chem. Phys., 16, 13945–13967, https://doi.org/10.5194/acp-16-13945-2016, 2016a.

Young, G., Jones, H. M., Darbyshire, E., Baustian, K. J., McQuaid, J. B., Bower, K. N., Connolly, P. J., Gallagher, M. W., and Choularton, T. W.: Size-segregated compositional analysis of aerosol particles collected in the European Arctic during the ACCACIA campaign, Atmos. Chem. Phys., 16, 4063–4079, doi:10.5194/acp-16-4063-2016, 2016b.

Young, G., Connolly, P. J., Jones, H. M., and Choularton, T. W.: Microphysical sensitivity of coupled springtime Arctic stratocumulus to modelled primary ice over the ice pack, marginal ice, and ocean, Atmos. Chem. Phys., 17, 4209–4227, https://doi.org/10.5194/acp-17-4209-2017, 2017.

Young, G., Connolly, P. J., Dearden, C., and Choularton, T. W.: Relating large-scale subsidence to convection development in Arctic mixed-phase marine stratocumulus, Atmos. Chem. Phys., 18, 1475-1494, https://doi.org/10.5194/acp-18-1475-2018, 2018.

---

## Referee Comment (RC2) · Anonymous Referee #2 · 3 Dec 2018

**Review of " Relaxation times of Arctic mixed phase clouds to short-term aerosol perturbations under different surface forcing" by Gesa K. Eirund, Anna Possner and Ulrike Lohmann**

This manuscript deals with the impact of short-term aerosol perturbations on Arctic mixed-phase clouds (MPCs). Based on an observed case from the ACCACIA campaign, the authors use large-eddy simulation to investigate how the structure and microphysical properties of the modelled MPC change with changing cloud condensation nuclei (CCN) and ice nucleating particle (INP) concentrations, changing surface fluxes (characteristic for a surface covered with ice or open water) and air temperature. The manuscript is in general well written and contains interesting results. However, there are a few points that I think should be clarified before the manuscript is accepted for publication.

**Major comments:**
1. Introduction: It is not clear to me *why* a short-term aerosol emission perturbation should generate a *long-term* cloud response. Since (accumulation mode) aerosols are efficiently scavenged by precipitation, I don't see a clear reason why the response would last longer than a few hours if you have precipitating clouds? I was actually quite surprised that you did indeed see a quite pronounced response up to almost a day after the perturbation. I think that a motivation would be good to include in the introduction.
2. Model description and setup: The model description and simulation setup is incomplete. I would suggest adding information on:
    a. Which hydrometeors are considered? Is it only one category for liquid and one for ice or do you have separate categories for cloud droplets, rain drops, pristine ice, graupel/hail, snow,….? And what is really $q_c$ and $q_i$?
    b. Are any secondary ice formation processes represented in the model?
    c. At which altitude is the model top?
    d. At which latitude is the case simulated?
    e. What is the time step and how long is the integration time?
    f. What is the assumed habit of the ice crystals?
    g. Is the sounding performed over ice or open water?
       Related to the last comment, I'm not convinced that it's appropriate to use the same sounding to initialize the "open water" and "ice surface" simulation. I would assume that a sounding over ice looks quite different to a sounding over water?
3. Evaluation of background state:
    a. The definition of the cloud extent is confusing (but it may become clearer if point 2a above is clarified). Does the limit of $q_c$ applied to define the cloud boundary include rain? In Section 3, you say that "Our model successfully simulates a liquid-topped MPC with ice sedimenting out of the liquid layer in both control simulations, according to observations". But this structure is not really clear from Figure 4. Looking at figure 4, I seems like you have a cloud between approximately 600 and 1500m (in the ocean case) and that the rest is falling precipitation?
       The cloud illustrated in Figure 12 is also quite different from the clouds plotted in Figure 4, the simulated cloud over the ice surface is not substantially thinner than the cloud over ocean (at least not according to the values in Table 2).
    b. Droplet number concentration: It is not clear to me how you could possibly get a droplet number concentration of 63 or 110 cm$^{-3}$ (as observed over ocean and ice,

respectively) in your simulations if you have a CCN concentration that is only 49 cm$^{-3}$. On the other hand, the CCN concentration plotted in Figure 2b is approximately 100 cm$^{-3}$ at ~2000 m. Why is the concentration so high at his altitude? This seems to be much higher than the cloud top, so I don't think it could be transported CCN?

4. Robustness to perturbations in microphysics: Figure 6 shows that the LWP increase with increasing CCN concentration is more sustained over ocean than over ice. The authors also discuss this result on page 11-12, but I cannot really find an explanation/hypothesis to why this is the case? And, as mentioned above in comment 1, I was actually surprised to see that the CCN perturbation response was sustained for such a long time after the initial perturbation – in particular for the open ocean case.

5. Discussion:

   a. The simulated transport of CCN out of the cloud layer is interesting, but also a bit puzzling to me. In another Arctic mixed-phase cloud study, Igel et al. (2017) found that entrainment/mixing *from* the free troposphere could actually be an efficient *source* of aerosols/CCN to the mixed-phase cloud layer. I think this would be worth mentioning/discussing. I'm not sure why you (and Solomon et al., 2018) get a different response, but one possibility could be that the cloud simulated by Igel et al. extended into the inversion, and that the inversion layer and lower free troposphere was actually moister than the below-cloud layer. Another possibility could be that Igel et al., impose a vertical gradient in their aerosol concentrations (based on observations). In general, I think it would be interesting to see how efficient the CCN recycling is in your study.

   b. Another thing (related to the above point) that would be interesting to know is how well COSMO-LES simulates the entrainment processes at the cloud top. Do you have any observations of TKE dissipation from ACCACIA that you could compare with?

   c. In general, the authors could extend and contrast their results to other studies on aerosol effects on mixed-phase clouds. The increase in ice water content and subsequent decrease in liquid water content with increasing INP has been described by e.g. Avramov and Harrington (2010), Ovchinnikov et al. (2014), Young et al. (2017), Stevens et al. (2018). The increase in liquid water path with increasing CCN has been discussed in Stevens et al. (2018)

**Minor comments:**

Abstract:

1. Although it's mentioned in the title, I think it should be clarified also in the abstract that you are looking at instantaneous aerosol perturbations.

2. The sentence starting with "Motivated by ongoing sea ice retreat…" does not read very well. It's not clear what you contrast with what and that it is model simulations you are referring to.

Introduction:

3. Page 2, lines 4-6: Deep convective clouds are also mixed-phase.

4. Page 2, line 10: "… potentially causing a warming effect…" – why potentially? Isn't the LW (surface) effect always warming?

5. Page 2, line 15: "… eventually accelerating…" – why accelerating? Isn't it also possible that we could have a negative feedback from clouds?

6. Page 2, lines 25-29: This sentence does not read very well.
7. Page 2, line 33: Why would sea salt and dimethyl emissions dominate ship emissions? Do you mean that these (sea salt and DMS) emissions generally are larger than ship emissions?

Model description and setup:

8. Page 3, line 30: "km" should be "km$^2$".

Evaluation of background state:

9. Page 5, line 3: I would specify that "both control simulations" refer to the control simulations over open water and ice, respectively (and thereby define *ocean_control* and *ice_control*).
10. Page 5, line 9: "… lower in our model simulations". Lower than what? I assume you mean compared to observations?
11. Figure 1: Perhaps refer to table 2 for the simulations?
12. Table 2: "Cloud extend" should be "cloud extent".

Surface flux impact on cloud dynamics:

13. Figure 3: What time step is plotted?
14. Page 6, line 5: "Cloud extend" should be "cloud extent".

Invariance of results across temperature regimes:

15. Page 13, line 10: I would suggest adding that the RH is kept constant, just as a clarification.

Consistent response independent of perturbation injection period:

16. Page 15, line 9: I would suggest adding "substantial" in between "no" and "change".

Discussion:

17. Page 16, line 15: What is $\tau$?
18. Page 17, line 4: Define WRF?
19. Page 19, line 15: I suggest changing "resembling" to "providing" or something similar.

**Reference**

Avramov, A. and Harrington, J. Y.: Influence of parameterized ice habit on simulated mixed phase Arctic clouds, J. Geophys. Res., 115, 1–14, https://doi.org/10.1029/2009JD012108, 2010.

Igel, A. L., A. M. L. Ekman, C. Leck, M. Tjernström, J. Savre, and J. Sedlar (2017), The free troposphere as a potential source of arctic boundary layer aerosol particles, Geophys. Res. Lett., 44, 7053–7060, doi:10.1002/ 2017GL073808.

Ovchinnikov, M., Ackerman, A. S., Avramov, A., Cheng, A., Fan, J., Fridlind, A. M., Ghan, S., Harrington, J., Hoose, C., Korolev, A., McFarquhar, G. M., Morrison, H., Paukert, M., Savre, J., Shipway, B. J., Shupe, M. D., Solomon, A., and Sulia, K.: Intercomparison of large-eddy simulations of Arctic mixed-phase clouds: Importance of ice size distribution assumptions, J. Adv. Model. Earth Sy., 6, 223–248, https://doi.org/10.1002/2013MS000282, 2014.

Young, G., Connolly, P. J., Jones, H. M., and Choularton, T. W.: Microphysical sensitivity of coupled springtime Arctic stratocumulus to modelled primary ice over the ice pack, marginal ice, and ocean, Atmos. Chem. Phys., 17, 4209–4227, https://doi.org/10.5194/acp-17-4209-2017, 2017.

---

## Author Comment (AC1) · 3 May 2019

Response to Reviewer #1:

The study titled "**Relaxation times of Arctic mixed-phase clouds to short-term aerosol perturbations under different surface forcings**" by Eirund et al. illustrates how mixed-phase clouds, modelled using large-eddy simulations, respond microphysically to short bursts of high aerosol number concentrations, such as those which would be experienced in the vicinity of shipping emissions. By simulating two cloud scenarios – one over sea ice, the other over ocean – the authors show how the surface conditions, moderated by chosen sensible and latent heat fluxes, can affect how the clouds respond to the influx of high aerosol particle number concentrations.

The study builds upon previous work using measurements from the Aerosol-Cloud Coupling And Climate Interactions in the Arctic (ACCACIA) campaign, and tests these observations with a more complex model representation of aerosol-cloud interactions than has been done before; therefore, the results are an important addition to the scientific literature. However, before publication, I have a few concerns which I feel should be addressed. The study has some potential implications for the stability of Arctic clouds in the face of increased shipping emissions across the region; however, these are not suitably discussed at present.

Furthermore, the authors come close to repeating some conclusions from Young et al., 2016a and 2017, and should distinguish the novelty of their results more so from these published works.

Thank you very much for the detailed review. We incorporated your suggestions within the revised manuscript, which substantially improve the quality of the manuscript. In response to some of the criticisms raised by both reviewers, we discovered two errors in our previously submitted model simulations that affected our results substantially. These two errors were found within the aerosol- and two-moment scheme, which induced an accumulation of aerosol above the cloud within the inversion layer. In more detail, the COSMO model has several clipping routines, where the cloud droplet number ($N_{drop}$) gets clipped, when cloud water content (qc) falls below a minimum threshold ($1 e^{-15}$ g m$^{-3}$). Within these clipping routines it is necessary to fill the cloud condensation nuclei (CCN) budget again with the amount of $N_{drop}$ that have been removed in order to conserve number. We generally included this adjustment in the code, but we were missing to include the CCN adjustment in one routine. Here, $N_{drop}$ was set to a minimum value (2 cm$^{-3}$) for qc > $1 e^{-9}$ g m$^{-3}$. This routine induced an artificial source of CCN above cloud top. Given the stable stratification in the inversion, the CCN released above cloud top were only re-entrained at very slow timescales into the cloud.  Additionally, we had to adjust the weights calculation that determines the redistribution of CCN after evaporative processes. Through a miscalculation in this routine, CCN were lost within the boundary layer, which impacted our total $N_{drop}$ throughout the simulation. As these two errors had compensating effects on the total CCN budget, we discovered them only once we had the first one fixed.

In the substantially revised manuscript we now focus more on the ocean/sea ice difference and the microphysical pathways of the cloud response to aerosol perturbations rather than the timescales of cloud adjustment. However, many of your comments remain valid, such that we included many your suggestions in the revised manuscript. Some sections of the previous manuscript were deleted/rewritten, which we clearly stated in the specific answers to your comments. Please find below our responses, which we marked in red.

**General comments:**

1. The paper does not suitably cite previous work on this case study from the ACCACIA campaign. The work is novel in its prognostic representation of both CCN and INP; however, similar studieshave been already conducted which compare cloud microphysics over sea ice and over ocean. Please ensure all previous literature is cited more appropriately. For example, the differences between the boundary layer structure over sea ice and over ocean has been discussed as an observational study by Young et al., 2016a, and large-eddy simulations of this case study have been presented by Young

et al., 2017. Observational conclusions should not be repeated here as conclusions of this study unless these earlier works are cited appropriately.

We thank the reviewer for pointing out that we were not sufficiently clear in distinguishing the novelty of this work from previous works. For a thorough comparison to work performed by Young et al., 2017, we included their LES simulation profiles (as shown in Figure 7 from their study) into our Figure 4 and Table 2.
Moreover, we emphasized more strongly in the text, which results were already obtained from observations and which ones are new.

2. The same boundary layer properties are used to compare how clouds form over the ocean or sea ice from the same state. Whilst this is an interesting perspective, why did the authors not use a boundary layer profile measured over the sea ice for these simulations? Do the authors expect the resulting cloud to compare well with observations when the initial profiles used are not the same as that measured? The boundary layer over sea ice is different to that over the ocean (as presented by Young et al., 2016a; 2017); therefore, can the authors comment on why they used the oceanic profile for the sea ice simulations?

Concerning the boundary layer profile over sea ice and ocean we included a more detailed explanation in the text.
Initially, we did use the sea ice dropsonde profile to initialize the modeled sea ice case, however, the profile was too dry to simulate a mixed-phase cloud for the given atmospheric state. Hence, we used the profile over the open ocean, which also provides a better comparison to the modeled open ocean case and reduces differences between the two cases solely down to differences in surface fluxes. We agree with the reviewer that a quantitative direct comparison of the sea ice case with the observations obtained over sea ice (as in Young et al., 2016) is not completely valid, since the initial state differs. However, we argue that a qualitative response (i.e. lower liquid and ice water content over sea ice than over the open ocean, higher cloud top and cloud base over the open ocean) is still valid and is consistent with observations from Young et al., 2016.

3. Readability and clarity could be improved – for example, it is often not clear whether the model simulation results or measurements are being discussed.

We improved clarity and readability and introduced a naming convention to distinguish more clearly between the observations and simulations (see Table 1).

4. The Discussion section could be significantly improved – it currently focuses on validating findings against previous studies; however, there is an opportunity to compare with previous ACCACIA studies which is currently not being capitalised upon. Specifically, there is an opportunity to conduct comparisons with the LES findings of Young et al., 2017 – your results are similar, and therefore there is scope to make some preliminary statements about the ability of two different models to reproduce these observations.

We included the comparison with results from Young et al., 2017 in the results section (see comment 1) and in the discussion.

5. The INP perturbation experiments are lacking analysis and discussion. Please add to this section of the study or remove it.

Thank you for pointing this out. We agree that the section discussing INP perturbations lacked more in-depth interpretation. We added more analysis to section 5.2 and additional discussion on the INP perturbation experiments in section 6. In particular, we included a summary of the mean cloud

properties for the simulations perturbed by INP in Table 3, as well the temperature sensitivity simulations for a perturbation of 10 INP L$^{-1}$ in the appendix and a panel demonstrating the effect of INP perturbations in our final schematic summary (Fig. 11).

6. The authors have the opportunity to make some preliminary statements about the stability and microphysical response of Arctic MPCs in the face of pollution transport/shipping emissions (as suggested in the Introduction); however, little discussion of this is included. Please comment on the potential real-world implications of this modelling study.

First, we added references for our chosen CCN and INP perturbations. Our chosen numbers are within the range of CCN observed in ship exhaust plumes (Hobbs et al., 2000) and Arctic haze (Rogers et al., 2001). Additionally, we added more discussion on a comparison to MPC satellite observations in ship tracks from Christensen et al., 2014 in section 5.1.
To make assumption about the future impact of sea ice loss and pollution in the Arctic (as stated in the Introduction), we included a statement at the end of our conclusions (page 22, line 30ff).

**Specific comments:**
**Abstract:**
**Page 1:**
**Line 4:** was ACCACIA conducted in the central Arctic? I would've taken this to be >80N?
Changed to "European Arctic".
**Line 5:** define COSMO
Defined.
**Lines 6–11:** these findings read very similarly to those presented by Young et al., 2016a; 2017. Are they conclusions from your modelling work? It currently reads like conclusions from the measurements used to initialise the model – measurements that have already been published. Additionally, LES studies of this case study have already been published. Please distinguish your conclusions more so from these studies to highlight its novelty.
We added that our obtained results agree with observations, to point out that these findings are rather a reproduction than a new result (page 1, lines 8-9).
**Line 12:** "*two dynamically different regimes*" – this is quite vague, can you expand on this?
We changed this line to "*sea ice and open ocean cloud regime*" (page 1, line 13).
**Line 16:** "*the maximum response*" – response in what?
No longer applicable. Entire abstract was rephrased.
**Line 18:** Could you specify more about your aerosol perturbations here? For example, their duration/altitude?
We added this information on page 1, line 13ff:
"*Perturbation aerosol concentrations relevant for CCN activation were increased to a range between 100 to 1000 cm$^{-3}$ and INP perturbations were once doubled 15 as compared to the background concentration and once increased by a factor of 3 (at every grid point and at all levels).*"
**Line 20:** Can you say more here about how the aerosols are transported out of the boundary layer?
Previously the aerosols had been accumulating in the inversion layer above cloud top, as explained in the first paragraph. However, this was due to a bug within the simulation, which has been resolved for the revised manuscript.
**1. Introduction:**
**Page 2:**
**Lines 13-15:** Please rephrase – sentence meaning unclear
Sentence changed to "*Due to their strong radiative impact, MPCs can alter the Arctic climate system (e.g. Bennartz et al., 2013; VanTricht et al., 2016), potentially accelerating or slowing the current high latitude warming*" (page 2, lines 15-16).
**Line 16:** "*amount*" is vague – "*fraction*"?
Changed to "*fraction*" (page 2, line 17).

**Line 23:** Please provide references for the sentence ending "*potential implications for cloud dynamics*".

We added references from the follow up section on cloud dynamics (Schweiger et al., 2008; Sotiropoulou et al., 2016; Young et al., 2016; Young et al., 2018).

**Line 23:** Please define ECMWF.

Defined.

**Line 29:** please rephrase

Rephrased to "*These observed changes in cloud height were also observed during the Aerosol-Cloud Coupling And Climate Interactions in the Arctic (ACCACIA) campaign (Young et al., 2016). Besides, the authors reported fewer and larger cloud droplets as well as increased precipitation rates over the open ocean as compared to over sea ice.*" (page 2, lines 28-31).

**Line 34:** Can the Arctic still be called pristine? It is cleaner than the mid-latitudes and is very clean in the summer, but the Arctic haze strongly increases aerosol mass concentrations in the Arctic atmosphere during the spring, where the data used to initialise these model simulations was collected. Please consider rephrasing this statement.

That is true, but even in spring the Arctic is still more pristine than other regions on Earth, especially the mid latitudes (Moore et al., 2013, Schmale et al., 2018). We added this clarification in the text (page 3, line 3).

**2. Model description and setup**

**Page 3:**

**Line 27:** Dropsondes were discussed in detail in Young et al., 2016a – please cite here for reference.

Included.

**Lines 29-31:** Spatial domain size and resolution are specified, please include similar information regarding run length and temporal resolution.

Included (page 4, lines 8-9).

**Page 4:**

**1st paragraph:** Solomon et al., 2018 (ACP) use the DeMott et al, 2010 (PNAS) parametrization, not the DeMott et al, 2015 (ACP) parametrization. They use a prognostic derivation of the temperature-dependent fit from Fig. 2 of DeMott et al., 2010, removing the dependence on aerosol number concentration from the standard version of the parametrization. Please can the authors clarify which ice nucleating particle parametrization they used, and note whether they included the aerosol number concentration dependence or used the version (described in Solomon et al., 2015, ACP) which is dependent only on temperature? DeMott et al., 2015 (ACP) is designed for mineral dusts; therefore, if this relationship was used, can the authors comment on the validity of doing so in the Arctic where there is some dust (Young et al., 2016b, ACP) externally mixed with other aerosol species?

We clarified in the text that for our study (as for Possner et al., 2017), the underline{implementation} is the same as in Solomon et al., 2015, but the curve to be discretized into the 15 temperature dependent INP bins follows the new DeMott parameterization (page 4, lines 15-17).

**Lines 7-8:** Are these median diameters that are quoted? Please clarify.

No, they are mean diameters. We included this in the text.

**Line 9:** "*low altitudes*" – please specify. Did the authors exclude in-cloud measurements with the PCASP (standard practice)? If so, how?

No, we did not use PCASP measurements within the revised manuscript.

With our fixed aerosol scheme we noted a fast depletion of CCN by precipitation in our *control* simulations, which led to an underestimation of the observed $N_{drop}$. To provide a better comparison to the observed cloud properties from Young et al., 2016, we kept the background CCN concentrations fixed at 100 cm$^{-3}$, which then leads to a more reasonable $N_{drop}$ (as compared to observations). Additionally, it matches the fixed $N_{drop}$ in the Young et al., 2017 LES simulations for

the same case, which we included for comparison (see General comment 1). Hence, we rewrote this section of the model description (page 4, lines 22ff).

**2nd paragraph:**
Please say more about how the aerosol observations are used as input to the model. The PCASP only measures >0.13 micron, do you have any other aerosol measurements for the smaller size mode? When fitting these two modes to the observations, what geometric standard deviation was used for each? Was a lognormal distribution assumed? Did you fit to data collected over the entire ocean segment of the ACCACIA flight in question, and did you use different inputs for the sea ice simulations?

As mentioned above, we used the constant $N_{drop}$ in Young et al., 2017 and the $N_{drop}$ from the observations as our new reference for choosing our initial CCN concentration (100 $cm^{-3}$). Hence, we don't mention PCASP measurements any more.

Generally, our CCN modes are represented as lognormal size distributions. We added the standard deviation in the explanations (page 4, lines 21-22). For simplification, the aerosol input profile is constant with height and we assumed the same aerosol background profile for the open ocean and the sea ice case. Similar to the thermodynamic background conditions, we wanted to keep everything except the surface conditions the same in both control simulations, to reduce differences to those in surface fluxes.

**Lines 10-11:** How did you arrive at this INP concentration? In Young et al., 2016a, the parametrizations listed are temperature dependent and were evaluated at the coldest temperature measured over the sea ice or ocean. Is the same technique used here? 3.3 $L_{-1}$ is higher than presented in Young et al., 2016a for PCASP data over both the sea ice and ocean (their Table 3), and seems particularly so given the sensitivity to ice particle number concentration presented by Young et al., 2017. Please provide more information about where this number concentration has come from.

We agree that the cloud properties are highly dependent on the INPs (as indicated by the INP sensitivity experiments and as shown in Young et al., 2017). Given the tremendous uncertainties with regard to secondary ice formation, the INP concentration was constrained to provide realistic INP and IWP within these simulations. Under these constraints we arrived a total INP concentration of 3.3 $L^{-1}$ distributed as prescribed by D10 over the 250.5 – 258 K temperature range. We note that this is higher than 0.5-1.5 $L^{-1}$ during ACCACIA and ISDAC, but we wanted to prevent a large underestimation of the ice phase in our simulations. With this initialization we are able to reproduce LES results from Young et al., 2017 using the ACC-ice parameterization (Table 2 and Fig. 4).

**Line 14-15:** How did you arrive at these values? They are within the range measured (Young et al., 2016a), yet they are very specific choices which should be justified.

We did some sensitivity analysis towards higher surface fluxes but we noted that with stronger surface fluxes, updraft speeds increased such to create stronger convective cells. For increased convection and stronger cloud organization the domain size should be increased. As the response to higher fluxes was linear and to save computational power without increasing the domain size even more we kept the fluxes at the lower end of the observations. We included this explanation in the text (page 4, lines 32-35).

**2.1 Model perturbation experiments**
**Line 22:** are all model analyses taken after 1.5h? Is this taken to be the spin-up period of the model? From Fig. 6, it looks like the spin up may be until 2h? Can the authors discuss whether other diagnostics (such as TKE or W) were used to define the spin up? Additionally, how was the perturbation time chosen? Was this taken to be immediately after spin up (inferred from Fig. 6)?

Spin-up was identified after the major precipitation event was over in the beginning of the simulation. We added the spin-up length to our model setup description on page 4, line 9. As we wanted to capture the full aerosol response and our simulation time was limited due to computational cost, we could not delay the aerosol injection further.

The perturbation injection was chosen to be directly after the initialization. For clarification we rephrased this in the text (page 5, lines 6-7).

**Lines 22-23:** Why is the pollution mode smaller? I agree it should be, but more could be done to justify this choice. Why was 0.19 micron specifically chosen? Again, is this the median diameter of the mode?

The pollution mode is motivated by anthropogenic particle compositions such as sulfates, nitrates, black carbon and organic carbon. These particles are on average smaller than natural aerosol sources such as sea salt and dust. With the pollution aerosol mode we wanted to include both, pollution from ship exhausts as well as transported aerosol from the mid-latitudes representing Arctic haze during spring (Law and Stohl, 2007). With a size of 0.19 micron we are still within the range of ship emissions (Hobbs et al., 2000) and include similar particles as the background mode.

As wanted to ensure the perturbation mode CCN to activate after the background CCN, its size needed to be slightly smaller due to its implementation in the code (as now included in the text, page 5, lines 9-10).

**Line 26:** This reads like you have both doubled the background and increased by a factor of 3 (i.e. ~16.6 L$_{-1}$) – please clarify

This has been clarified in the text (page 5, line 15).

**Line 28:** Please provide references for this statement.

We added a reference (Devasthale and Thomas, 2012).

**Line 21:** How long is a single time step? (see previous comment on temporal resolution)

One single model time step is 2 s, this has been added in the model description (page 4, line 9).

**3. Evaluation of background state**

**Page 5:**

**1$_{st}$ paragraph:** Here the authors seem to jump between the simulations and the initial conditions and discuss these interchangeably. Please describe the initial conditions (dropsonde measurements) first, then the controls to avoid confusion.

We changed the paragraph by first describing the observations as reported in Young et al., 2016 and then describing the simulated profiles shown in Figure 1.

**Line 9:** *ocean_control* is used interchangeably to refer to the observations and the control model simulations.

This label should refer to model results only. Please use a different label (e.g. *observations*) to describe the dropsonde measurements. Also, it lists the mean N$_{ice}$ is 0.17 L$_{-1}$ in Table 2?

As you suggested, we use the label *observations* to describe the observed characteristics form Young et al., 2016 to avoid confusion.

Thanks for pointing that out, that was a previous typo in the text. As our simulations changed due to the new model setup, we adjusted all values in Table 2 and the text accordingly.

**Lines 11-13:** Did you try changing the background concentration of CCN to improve the cloud microphysical agreement with observations? 3.39/3.99 cm$_{-3}$ is significantly less than observed – are these comparisons with the observations robust? There are significant differences in cloud base/top height and cloud properties, the study would therefore benefit from some discussion on why this is the case and how these differences may affect the real-world implications.

This underestimation was largely due to a bug within our simulations, which induced an unrealistic vertical distribution of aerosol and low CCN concentrations within the cloud and sub-cloud layer. This has now been fixed. To further avoid a slow decline in CCN throughout the simulated period due to losses by surface precipitation, the background CCN were now held constant. This section is updated accordingly.

**Line 14:** Why do those aerosols in the inversion layer not activate? Is it sub-saturated? Is there too much competition for water vapour?

Yes, the air above cloud was sub-saturated and the entrainment into the cloud occurred on very slow timescales. However, this no longer applies to the bug-fixed simulations as the strong aerosol accumulation above cloud top is resolved.

**Line 15:** Please provide a comment on the realism of this finding (poor re-entrainment of aerosols due to low turbulence).

Also this is resolved with the new setup.

**Lines 15-17:** Please clarify. Are you talking about the dry run where you have $N_{drop}$ formation? How is this possible if the boundary layer is kept below water saturation?

Resolved with new setup/statement deleted.

**Page 6:**

**Line 1:** Do you show this anywhere? (Mixing of CCN from above)? Or is it inferred from Fig. 2? Perhaps some w' tendencies could be shown to illustrate upwards/downwards motion (if these diagnostics are available)?

Resolved with new setup/statement deleted.

**4. Surface flux impact on cloud dynamics**

**Page 7:**

**Lines 1-2:** Young et al., 2018 discuss similar simulated effects from oceanic surface fluxes using these observations for initialisation – please consider cite/comparing here.

As we compare similarities/differences to other studies in the discussion, we added a comparison to Young et al., 2018 in the discussion (page 17, lines 29-31). Thank you for the additional reference. These results agree nicely with our open ocean case.

**5.1 Response to CCN perturbations**

**Page 9:**

**Line 6:** "*is sufficient to significantly perturb*" – please elaborate. By how much? Do smaller perturbations not change the cloud physics as much? There is an opportunity to discuss real-world consequences here.

For the following Figures 7,9 and 10 (showing the LWP and IWP) we changed to showing the mean plus/minus one standard deviation instead of the median. Also, with the new simulations our figures have changed substantially.

Regarding your comment, we now state in the text that a perturbation of 100 cm$^{-3}$ CCN is sufficient to perturb the mean LWP by 13% within the first hour upon seeding. To include some reference to observations, we compared that to estimates of Christensen et al., 2014, who found an equivalent change of LWP in ship tracks (page 11, lines 9-10).

**Line 15:** For reference to Fig. 6C to be valid, need to mention IWP here too.

As we rewrote this paragraph the reference to IWP has been moved. We mention IWP and the respective reference to Fig. 7c now on page 11, line 7.

**Page 10:**

**Line 7:** Young et al., 2018 showed that these detraining layers of moisture can be reduced by implementing strong large-scale subsidence. Cloud top increases with time in your ocean simulations due to the heat and moisture fluxes from below – have you looked at the effect of increasing your imposed subsidence to reduce cloud deepening?

No, unfortunately we did not perform any sensitivity studies concerning subsidence rates. However, we plan to investigate the influence of large-scale variability (inversion strength which might impact aerosol detrainment, subsidence, boundary layer stability etc.) in a separate study.

**Lines 10-11:** Why? Opportunity to discuss.

The section comparing the response to CCN perturbations over the open ocean and sea ice has been rewritten/moved to page 13, line 16ff. But the general statement that the response over sea ice lasts longer remains still valid, which is discussed now on page 13, lines 26ff.

**Line 14:** "*most perturbed simulation*" – please quote run label for clarity

Generally, we use the simulation names as stated in Table 1 now for reference to the individual simulations.

**Page 11:**

**Line 4:** "*increase in the ice phase*" – please be more specific, do you mean number concentration? Mass concentration?

In the rewritten paragraph we now refer to "$N_{ice} \ and \ IWP$" on page 13, line 1.

**Lines 5-6:** Even though the largest perturbation simulation LWP relaxes back to similar trends as the control simulation, it's magnitude is still approximately 2-3× that of the control. Given the low LWPs

simulated, this small difference could have an effect on the radiative properties of the clouds. Please discuss.

This has changed now with the new simulations. These statements in the text have been deleted.

**Page 12:**

**Line 1:** Figure 2b does not show transport out of the boundary layer, it just shows non-zero number concentrations. To prove transport, could you show some tendencies (perhaps some relationship with w')? Or perhaps a time series of aerosol particle number concentration profiles (like Figures S2-S4)?

This has changed now with the new simulations. These statements in the text have been deleted.

**5.2 Response to INP perturbations**

**Page 13:**

**1st paragraph:** Could the authors show how the INP perturbations affect $N_{ice}$, in addition to LWP/IWP? Also, there is little analysis on this section's findings in comparison to the CCN perturbations, why is this? As the manuscript stands, this section reads like an afterthought.

We agree that the discussion of the INP perturbations was insufficient in our manuscript and added more analysis/discussion. We added the cloud properties as obtained from the *10INP* simulations in our Table 3 (which includes $N_{ice}$).

**Lines 6-7:** Is this illustrated anywhere? If not, please include a figure (like Figure 7) in the supplementary as evidence.

This has been rewritten to "*the stratus cloud over sea ice is initially very susceptible to INP perturbations, which induce an initial peak in IWP and surface precipitation before the cloud returns to the unperturbed state*" (page 16, lines 3-4). For reference, we added a figure showing surface precipitation in the appendix, Fig. S1c,d.

**Side note for Discussion:** The authors show that the cloud does not glaciate (in agreement with other studies). These findings are in contrast to Young et al., 2017's ACCACIA LES results, who use a more simplified representation of cloud microphysics and aerosol-cloud interactions. Could this mixed-phase persistence be because the ice number concentrations are much lower than observed, and modelled in that case? There is an opportunity to compare with their findings, which the authors do not capitalise on. Also, there is a lack of analysis/discussion on the INP perturbation experiments – does the extra ice created by the INP injection precipitate out of the cloud as snow? If so, how does the INP injection affect precipitation rates?

Thank you for this additional discussion point. We included a comparison to Young et al., 2017. As is our new simulations $N_{ice}$ is close to what has been modeled by Young et al., 2017, we don't think that $N_{ice}$ is responsible for the lack of glaciation in our simulated case. As $N_{drop} \gg N_{ice}$ in our simulations (see our Table 3) and also in simulations from Young et al., 2017, we generally think a complete glaciation of the cloud in Young et al., 2017 is very surprising.

We added this as well as a more in-depth comparison to the impact of CCN perturbations in section 5.2 as well as in the discussion.

**5.3 Invariance of results across temperature regimes**

**Page 14:**

**Lines 6-10:** Please improve clarity

The wording in this paragraph has been changed to improve clarity.

**Line 12:** Again, is this the parametrization used? This is not the same as in Solomon et al., 2015.

We clarified this already in the model description section (see comment above, only the implementation is the same as in Solomon et al., 2015, but the newer DeMott parameterization (DeMott 2015) is used).

**5.4 Consistent response independent of perturbation injection period**

Note: this section has been completely removed, as the cloud response to CCN perturbations has been changed with the fixes in the aerosol scheme.

**Page 15:**

**Lines 2-4:** This should be made clearer at the start – to me, it was not clear until now what the aerosol perturbations represented in model terms.

**Side note:** This section seems to be "in response" to some discussion, perhaps it should be relocated to a sub-section of Section 6?

**6. Discussion:**

**Page 16:**

**Line 5:** Reference to Young et al., 2016a – this study uses a range of aircraft measurements to show this, and model results here should be presented as successfully reproducing these conditions rather than new conclusions.

Yes, we included a reference and comparison to Young et al., 2016.

**Line 11:** As previous, did you try using a sea ice dropsonde profile? Like that presented in Young et al., 2017 for this case? These conclusions are very similar to the observational conclusions of Young et al., 2016a and modelling conclusions of Young et al., 2017. Please reference these studies here – there is a great opportunity to show how these results compare with the previous studies, especially since a more complex microphysical modelling representation is used here. There is novelty in these results; however, the distinction between conclusions from this study and those from previous ACCACIA work is not clear.

As explained previously we tried but finally did not use the sea ice profile.
We also added a comparison to Young et al., 2017, as we also included their data into our results section (Table 2 and Figure 4).

**Line 13:** "*possible pathway for cloud-aerosol interactions*" – what is meant by this statement? Not clear/vague.

We changed this phase to "*As our model setup allows for a prognostic treatment of aerosol-cloud interactions, we are able to quantify the cloud response to spatio-temporally resolved aerosol perturbations* […]" (page 18, lines 10-12).

**Line 15:** Define $\tau$

Throughout the manuscript this has been changed to "cloud optical depth" to avoid confusion.

**4th parapraph:** The authors refer to "*polluted*"/"*unpolluted*" – are you referring to the CCN perturbation experiments only? There is no reference to the INP perturbation experiments here, and there is a lack of analysis/discussion on these simulations.

The focus on polluted/unpolluted periods has been removed in the revised manuscript (see our first comment on page 1 of the response). The quoted values in Table 3 and the text are now averages over the full time period. Additionally, we included the *10INP* simulations.

**Lines 32-33:** It has been previously stated that the perturbation experiments relaxed back to their initial state but the authors have here clarified that there is some difference in magnitude (as per my previous point). This is confirmed in the values quoted in Table 3 between the controls and "post-polluted" rows. Please ensure analysis and discussion is consistent throughout the manuscript.

This has been removed, as we deleted the division into a polluted and post-polluted period.

**Page 17:**

**Line 2:** Do you show aerosol particle transport out of the boundary layer? It is possible I have missed it, but non-zero values above don't necessarily show that aerosols are being transported vertically. Perhaps some microphysical tendencies (if you have the relevant diagnostics) could show the upward transport of aerosols?

Or a time evolution of aerosol number concentration (like Figures S2-S4)? As it stands, this statement does not seem to be supported by any figure in the manuscript or supporting information.

With the new set of simulations and the fixes to the scheme this has been deleted/become redundant.

**Lines 5-7:** There are more caveats to this study than listed here. For example, the fact that a sea ice boundary layer profile was not used for the sea ice simulations is a significant caveat that requires discussion. Why was this not used?

We do not consider the use of the ocean profile for both cases (sea ice and open ocean) a caveat of this study, as we now (as already explained previously) can clearly attribute differences between the two cases to differences in surface fluxes. In the case of freshly formed sea ice, reduced surface fluxes may impact the overlying clouds. Similarly, surface fluxes increase over polynyas which is also likely to impact clouds (Gultepe et al., 2003).

We agree that the results might not be valid for a large sea ice covered domain, as there the initial conditions might be different. But also here, season, local conditions, location etc. play a large role as well. We added the point of sea ice observations/model comparison due to the different initial profiles, but as mentioned we don't consider this to be a shortcoming of this study.

Note that this section has been moved to the end of the study, following the conclusions.

**7. Conclusions:**

**Page 19:**

**Line 2:** References for "*… the subject of a number of recent studies*"

We added some references (page 22, lines 2-3).

**Lines 8-10:** Opportunity to link with Young et al., 2018 ACCACIA study (cumuli tower development – intermodal agreement)

We added this link to the discussion and added a reference to Young et al., 2016, 2017, 2018 (page 17, first paragraph of section 6) as well as to the conclusions (page 22, line 10).

**Lines 8-14:** Make stronger links to previous ACCACIA studies and make novelty of results more distinct from previously published conclusions.

See our comment above, we added "*Our simulations support previous results obtained for the ACCACIA campaign (Young et al., 2016, 2017, 2018)*" to this section (page 22, line 9-10).

**Line 18:** "*Over sea ice, cloud droplet growth is less efficient...*" – why? There has been little discussion of why microphysical processes occur differently over sea ice and ocean.

This statement has been changed to "*This increased moisture flux leads to an increase in the cloud LWP and IWP, larger cloud droplets and ice crystals and a higher cloud base and cloud top*" (page 22, lines 12-13). As mentioned, we relate the (slightly) larger clod droplets to an increased vertical moisture flux over the ocean through cumuli towers feeding moisture into the stratus layer. Is now also stated in point 1 in the conclusions.

**Lines 25-26:** How? Please provide details.

We removed this statement from the conclusions, as it is not one of the main points of this study but rather a further sensitivity experiment.

**Line 28:** "*possible pathway for aerosol-cloud interactions*" – this statement has been used before and the meaning is not clear. Do the authors just mean that aerosol plumes may affect cloud structure in the Arctic? Please clarify.

This statement has been deleted from the revised manuscript.

**Technical corrections:**

**Page 1, line 19:** typo → "**properties**"

Thank you for pointing this out, but this has now been deleted in the revised manuscript.

**Page 2, line 2:** "high **model** uncertainties"

Here, we refer to general uncertainties is cloud physics. Hence, we left the current wording.

**Page 2, lines 12**: "**mid-latitude**"

Changed.

**Page 4, line 23:** "to be **at** slightly smaller **sizes** than"

Changed.

**Page 4, line 25:** "successively" → "**in stages**"

This has been reworded to "*Perturbation aerosol concentrations relevant for CCN activation were increased by 100, 200, 500 and 1000 $cm^{-3}$.*", so explicitly state the concentrations of the perturbations (page 5, line 13)

**Page 5, line 8:** "according to" → "**agreeing with**"

Changed to "*in agreement with*" (page 6, line 12).

**Page 5, line 15:** "In a dry run,…" – new paragraph?

This has been deleted in the revised manuscript.

**Page 7, line 7:** "allow the **cloud** droplets as well as the **ice** crystals"

This has been slightly rewritten in the revised manuscript, but we changed "droplets" to "cloud droplets" (page 8, lines 6-7).

**Page 7, line 15:** "LW cooling is increased up to… " → "**LW cooling increases up to**… " – the former reads like you are modifying the LW cooling, not a simulated effect.
This section has also been rewritten in the revised manuscript, but thanks for pointing this fact out. This has been changed to "[…] *the higher liquid water content in the air column increases the cloud longwave (LW) emissivity*" (page 9, line 8).
**Page 7, lines 15-16:** "The more numerous ice crystals" → "**Higher concentration of** ice crystals"
This has been deleted in the revised manuscript.
**Page 8, Figure 4 caption:** typo → "**interquartile**"
This has been removed as we changed to mean/standard deviation.
**Page 9, line 11:** "upon seeding" → "**after seeing**"
With "*upon seeding*" we refer to the time past the aerosol injection. For clarity, we changed it to "*after seeding*" (note that this paragraph has also been rewritten, but the relevant wording can be found on page 11, line 8 and page 12, line 2).
**Page 10, line 8:** "The initial… on the cloud regime" – remove, vague and not required.
We use this sentence as a transition from the impact of CCN perturbations on the open ocean and the sea ice case. Hence, we decided to keep it, but we reworded to "*The response to CCN perturbations strongly depends on the cloud regime*" (page 13, line 16).
**Page 10, line 9:** "cloud droplet growth **is** limited"
This has been removed in the revised manuscript.
**Page 19, line 19:** should this be a new paragraph?
We added a new pullet point for the INP conclusions.

**References:** Some references are incomplete or incorrect.
We went through the references and corrected everything that was incorrect.

**Figures and Tables:**
**Table 1:** Please list columns as "Background CCN/INP".
We deleted this column, as it is the same number for all simulations.
**Table 2:** Total values taken over how long? The entire run? Excluding spin up? Please clarify.
- Caption: "*Note that the airplane did not sample the lower and upper levels*" – of what? The model domain? Please clarify/rephrase
We removed this restriction in the analysis and accordingly also in the text. As our maximum liquid water mixing ratio in Figure 4 is at a slightly higher altitude (at approx. 1.55 km height) than in the observations, we didn't want to exclude this maximum from the analysis by restricting the averaging to heights <1.5 km.
- in-cloud criteria: both the liquid and ice mass thresholds? Or just one or the other? Please clarify, and define $q_c$ and $q_i$
We defined qc and qi in the caption. We used the qc threshold for the cloud liquid properties and the qi threshold for the cloud ice properties. This explanation is added in the caption.
- Can you comment on the very low cloud base height with comparison to the observations over the ocean? Or the cloud top height which is almost double the altitude of that observed over the sea ice?
We removed the column with cloud extent values, but refer in the text to Figure 4. Our cloud extent was determined only based on Qc (no rain, only cloud water). However, we agree that the report of mean values here is confusing. Instead, we calculated the cloud base and top for the stratiform cloud layer, i.e. the layer where 80% of the domain grid points are cloud-covered. The reason our previously calculated cloud base was so low, was because we sampled the updraft cores that are spatially limited, but regions of high qc. Now, we added horizontal lines in Figure 4 to indicate the cloud extent of the stratiform cloud.

**Please increase legend size on all figures.**
Changed.

**Figure 1:**
- please choose different colours – hard to read
- improve readability – perhaps split into 4 panels? Over ocean/sea ice?

We split the figures into 4 panels to improve readability. Thus we could also decrease the number of lines and colors.

**Figure 2:**
- How do these profiles compare with observations?

We added a comparison of the simulated $N_{drop}$ with observations over the ocean in the text.
- Would it be clearer to have: ice control (black), ice perturb (grey), ocean control (red), ocean perturb (orange)?

To improve readability, we changes the figures to a) ocean case and b) sea ice case and plotted $N_{drop}$ and CCN in the same figure, such that these two quantities can be directly compared.

**Figure 3:**
- There is no increase in ice number with decreasing altitude like in the observations (Young et al., 2016a ACP), please comment on this. Similarly, for the LWMR – these trends are in contrast to those observed, please comment on why.

Over the ocean, we reproduce model results of Young et al., 2017 very well. The increase in $N_{ice}$ in the observations in Young et al., 2016 are most likely related to a shattering event and hence not physical.
- In caption, define LWMR

Done.

**Figure 5:**
- Why are the panels not shown to 0 m? Or at least to cloud base?

Because we wanted to restrict ourselves to the temperature range (T<258 K) where freezing occurs (at cloud base temperatures are too warm for immersion freezing as parameterized in our model). We added this explanation in the figure caption.

**Figure 6:**
- it may be beneficial to show Fig. S4 as additional sub-panels of Fig. 6 to show how the cloud structure evolves with time in the different scenarios

We stayed with only showing Fig. 6 (now Fig. 7), as otherwise the whole figure would be too crowded with more than 4 subpanels. We included a figure in the appendix showing the cloud top for the different sensitivity simulations. The Hovmoeller plots shown in the previous appendix are replaced, as we find our new figure showing the cloud top being of equal importance.

**Figure 7:**
- Just a side note, this figure does not print well (not clear which line is which). Consider changing colours used, or splitting into sea ice/ocean sub-panels?

We removed this figure and included Figure 6, which shows the depositional growth in a Hovmoeller plot. We note that depositional growth is generally most important to contribute to an IWP increase, hence we switched these figures.

**Figure 10:**
- This figure is particularly crowded and individual traces are hard to distinguish. Perhaps separate into further sub-panels? (e.g. ocean+1000CCN, ocean control, ice+1000CCN, ice control)?

  We kept this figure, but changed the color of the *1000CCN* simulation to orange, to keep it consistent with Figure 7 and increase readability.

**Figure 12:**
- Is precipitation always as rain? Again, do you refer solely to the CCN perturbation experiments for "polluted" case analysis. Please define what you mean by "polluted", and include some analysis on the INP perturbation experiments (or remove).

Precipitation contains rain and snow, thus we added snow in the schematic. As mentioned earlier, we removed a distinction between "polluted" and "unpolluted" and now included a schematic showing the impact of INP perturbations.

---

## Author Comment (AC2) · 3 May 2019

**Review of " Relaxation times of Arctic mixed phase clouds to short-term aerosol perturbations under different surface forcing" by Gesa K. Eirund, Anna Possner and Ulrike Lohmann**

This manuscript deals with the impact of short-term aerosol perturbations on Arctic mixed-phase clouds (MPCs). Based on an observed case from the ACCACIA campaign, the authors use large eddy simulation to investigate how the structure and microphysical properties of the modelled MPC change with changing cloud condensation nuclei (CCN) and ice nucleating particle (INP) concentrations, changing surface fluxes (characteristic for a surface covered with ice or open water) and air temperature. The manuscript is in general well written and contains interesting results.

However, there are a few points that I think should be clarified before the manuscript is accepted for publication.

Thank you very much for the detailed review. We incorporated your suggestions within the revised manuscript, which substantially improve the quality of the manuscript. In response to some of the criticisms raised by both reviewers, we discovered two errors in our previously submitted model simulations that affected our results substantially. These two errors were found within the aerosol- and two-moment scheme, which induced an accumulation of aerosol above the cloud within the inversion layer. In more detail, the COSMO model has several clipping routines, where the cloud droplet number ($N_{drop}$) gets clipped, when cloud water content (qc) falls below a minimum threshold ($1\,e^{-15}$ g m$^{-3}$). Within these clipping routines it is necessary to fill the cloud condensation nuclei (CCN) budget again with the amount of $N_{drop}$ that have been removed in order to conserve number. We generally included this adjustment in the code, but we were missing to include the CCN adjustment in one routine. Here, $N_{drop}$ was set to a minimum value (2 cm$^{-3}$) for qc > $1\,e^{-9}$ g m$^{-3}$. This routine induced an artificial source of CCN above cloud top. Given the stable stratification in the inversion, the CCN released above cloud top were only re-entrained at very slow timescales into the cloud. Additionally, we had to adjust the weights calculation that determines the redistribution of CCN after evaporative processes. Through a miscalculation in this routine, CCN were lost within the boundary layer, which impacted our total $N_{drop}$ throughout the simulation. As these two errors had compensating effects on the total CCN budget, we discovered them only once we had the first one fixed.

In the substantially revised manuscript we now focus more on the ocean/sea ice difference and the microphysical pathways of the cloud response to aerosol perturbations rather than the timescales of cloud adjustment. However, many of your comments remain valid, such that we included many your suggestions in the revised manuscript. Some sections of the previous manuscript were deleted/rewritten, which we clearly stated in the specific answers to your comments. Please find below our responses, which we marked in red.

**Major comments:**

1. Introduction: It is not clear to me *why* a short-term aerosol emission perturbation should generate a *long-term* cloud response. Since (accumulation mode) aerosols are efficiently scavenged by precipitation, I don't see a clear reason why the response would last longer than a few hours if you have precipitating clouds? I was actually quite surprised that you did indeed see a quite pronounced response up to almost a day after the perturbation. I think that a motivation would be good to include in the introduction.

Note that we changed the focus of the revised study away from adjustment timescales and the MPC response times to aerosol perturbations, but moved towards the differences in aerosol response between the open ocean and sea ice regime as well as microphysical pathways in clouds as a response to aerosol perturbations.

Nevertheless, we did expect that with an aerosol perturbation of 1000 cm$^{-3}$, which is 10 times more CCN than the observed value, we would simulate a strong response (i.e. eventually including complete suppression of rain), especially considering the fact that the simulated precipitation (especially over sea ice) was very low (0.59 mm d$^{-1}$ over sea ice and 1.12 mm d$^{-1}$ over the open ocean). Therefore, aerosol concentrations in the boundary layer are depleted over very long time-scales in these simulations and concentrations remain high after seeding throughout the simulated period.

Also, the surface fluxes and higher updrafts over the open ocean provide a source of moisture and high relative humidity, which favors new droplet activation as response to an aerosol perturbation. Hence, we were not surprised to see a strong sensitivity (especially initially) to aerosol perturbations.

According to the shifted focus of our revised manuscript however, we removed the timescale aspect from the introduction completely.

2. Model description and setup: The model description and simulation setup is incomplete. I would suggest adding information on:

a. Which hydrometeors are considered? Is it only one category for liquid and one for ice or do you have separate categories for cloud droplets, rain drops, pristine ice, graupel/hail, snow,....? And what is really $q_c$ and $q_i$?

b. Are any secondary ice formation processes represented in the model?

c. At which altitude is the model top?

d. At which latitude is the case simulated?

e. What is the time step and how long is the integration time?

f. What is the assumed habit of the ice crystals?

g. Is the sounding performed over ice or open water?

Related to the last comment, I'm not convinced that it's appropriate to use the same sounding to initialize the "open water" and "ice surface" simulation. I would assume that a sounding over ice looks quite different to a sounding over water?

We added the following information to the model description (section 2):

a. Hydrometeors: 5 hydrometeor types (cloud droplets, raindrops, cloud ice, snow, graupel) represented as gamma distributions with prescribed shape parameters and prognosed bulk mass and number concentrations

b. In our simulations, only the HP-mechanism is included, other secondary ice processes are omitted. However, due to the very cold cloud temperatures (-15 to -20 °C), secondary ice processes do not play a strong role (Hallett and Mossop, 1974).

c. 23 km model top

d. A box of 20 x 20 km$^2$ around the position of dropsonde 5 release (75°N, 24,5°E)

e. Model time step 2 s with model output every 6 minutes, integration time 20h

f. Snowflakes and ice crystals are assumed to be dendrites

g. We used only the dropsonde over open ocean. Initially we tried to set up the sea ice case with the dropsonde over sea ice, however, the dropsonde profile over sea ice was too dry to simulate a cloud for our setup.
We agree that over sea ice the boundary layer profile might highly differ from the boundary layer over open ocean (Young et al., 2016). But by having the same initial conditions for the open ocean and the sea ice case we can narrow any differences in cloud dynamics and microphysics down to surface fluxes, which become increasingly important over freshly melted sea ice or polynyas (Gultepe et al., 2003). This is now more clearly stated in the manuscript.

3. Evaluation of background state:
a. The definition of the cloud extent is confusing (but it may become clearer if point 2a above is clarified). Does the limit of $q_c$ applied to define the cloud boundary include rain? In Section 3, you say that "Our model successfully simulates a liquid topped MPC with ice sedimenting out of the liquid layer in both control simulations, according to observations". But this structure is not really clear from Figure 4. Looking at figure 4, I seems like you have a cloud between approximately 600 and 1500 m (in the ocean case) and that the rest is falling precipitation?
The cloud illustrated in Figure 12 is also quite different from the clouds plotted in Figure 4, the simulated cloud over the ice surface is not substantially thinner than the cloud over ocean (at least not according to the values in Table 2).

Our cloud extent was determined only based on cloud water content (qc), i.e. no rain. However, we agree that the report of mean values here is confusing. As the cloud over the open ocean features convective structures, the calculation of the cloud extent based on the highest/lowest level with qc>0.01 g m$^{-3}$ leads to a sampling of the updraft towers, which distorts the cloud extent of the main stratus layer.
Instead we now calculated the cloud extent of the stratiform cloud layer, which is defined as the layer where 80% of the domain grid points have qc>0.01 g m$^{-3}$. We included the stratiform cloud base and top as horizontal, dashed lines in Figure 4. Thus, everything below and above this line represents convective structures. Note that the black line and shading in Figure 4 is only cloud water mixing ratio (excluding rain mixing ratio), which we also stated more clearly in the figure caption of Figure 4. In our revised manuscript we removed the column with cloud extent values in Table 2 to avoid confusion, but refer in the Text to Figure 4.

Thanks very much for pointing out that deficit in Figure 12, we adjusted the figure (now Figure 11 in the revised manuscript) to better fit our findings from sections 3 and 4.

b. Droplet number concentration: It is not clear to me how you could possibly get a droplet number concentration of 63 or 110 cm-3 (as observed over ocean and ice, respectively) in your simulations if you have a CCN concentration that is only 49 cm-3. On the other hand, the CCN concentration plotted in Figure 2b is approximately 100 cm-3 at ~2000 m. Why is the concentration so high at this altitude? This seems to be much higher than the cloud top, so I don't think it could be transported CCN?

That is indeed true. This very strong aerosol accumulation at the cloud top in our previous Fig. 2b resulted from two errors in our aerosol scheme (as described in the first paragraph). As we changed our setup in the revised manuscript, we initialized our new simulations with a fixed CCN background concentration of 100 cm$^{-3}$. This equals the fixed $N_{drop}$ concentration in LES simulations for the same case from Young et al., 2017 and ensures a more accurate representation of $N_{drop}$ as has been observed during the campaign in Young et al., 2016. Also, we kept the background CCN fixed over time (as in Possner et al., 2017) to prevent a loss of background aerosols by precipitation and freezing. With this new setup we are able to reproduce the observed $N_{drop}$ better than in our previous setup with 48±15 cm$^{-3}$ as compared to 63±30 cm$^{-3}$ in the observations (see Table 2).

4. Robustness to perturbations in microphysics: Figure 6 shows that the LWP increase with increasing CCN concentration is more sustained over ocean than over ice. The authors also discuss this result on page 11-12, but I cannot really find an explanation/hypothesis to why

this is the case? And, as mentioned above in comment 1, I was actually surprised to see that the CCN perturbation response was sustained for such a long time after the initial perturbation – in particular for the open ocean case.

Over open ocean updrafts are stronger and the vertical moisture flux is increased as compared to the cloud regime over sea ice. This allows for fast additional cloud droplet activation and formation, especially in the updraft cores (see Figure S2). We included more discussion on these differences between the cloud regime over the open ocean and sea ice in section 5.1 (page 13, lines 16ff) and in the discussion (page 17, first paragraph).
Resulting from the fast increase of $N_{drop}$ over the ocean and the initial precipitation suppression (Fig. S1a), the response of the liquid phase in the cloud over the ocean can be maintained for approximately 16 h, until the response shifts from the liquid to the ice phase through increased ice crystal growth by deposition (Fig. 6c and Fig. 7c).
Over sea ice, total surface precipitation is very low (as mentioned earlier; Fig. S1b) and the ice water path (IWP) is substantially lower as compared to over the ocean (Fig. 7d), such that CCN can perturb the cloud even beyond 20 h.

5. Discussion:
a. The simulated transport of CCN out of the cloud layer is interesting, but also a bit puzzling to me. In another Arctic mixed-phase cloud study, Igel et al. (2017) found that entrainment/mixing *from* the free troposphere could actually be an efficient *source* of aerosols/CCN to the mixed-phase cloud layer. I think this would be worth mentioning/discussing. I'm not sure why you (and Solomon et al., 2018) get a different response, but one possibility could be that the cloud simulated by Igel et al. extended into the inversion, and that the inversion layer and lower free troposphere was actually moister than the below-cloud layer. Another possibility could be that Igel et al., impose a vertical gradient in their aerosol concentrations (based on observations). In general, I think it would be interesting to see how efficient the CCN recycling is in your study.

This aspect has been removed from our revised manuscript, as the aerosol accumulation at cloud top was due to bugs in our aerosol scheme.
As a side note, prior to fixing the model, we did some tests with a varied inversion strength which had little to no effect on that strong aerosol sink at cloud top. However, we plan to investigate the impact of boundary layer stability on aerosol-cloud interactions and cloud dynamics in a future study.

b. Another thing (related to the above point) that would be interesting to know is how well COSMO-LES simulates the entrainment processes at the cloud top. Do you have any observations of TKE dissipation from ACCACIA that you could compare with?

This point also becomes redundant in our revised manuscript, as we don't discuss aerosol transport any more.

c. In general, the authors could extend and contrast their results to other studies on aerosol effects on mixed-phase clouds. The increase in ice water content and subsequent decrease in liquid water content with increasing INP has been described by e.g. Avramov and Harrington (2010), Ovchinnikov et al. (2014), Young et al. (2017), Stevens et al. (2018). The increase in liquid water path with increasing CCN has been discussed in Stevens et al. (2018)

Thanks for these additional references. We extended the comparison of our results to previous ACCACIA work and further studies on aerosol effects on Arctic MPCs.
More specifically, we added a comparison to LES results from Young et al., 2017 for the same case in section 3 as well as in the discussion.
Additionally, we expanded our discussion on INP perturbations and included references additional in section 5.2 and the discussion (e.g., Morrison et al., 2008, Ovchinnikov et al., 2014, Possner et al., 2017, Young et al., 2017, Solomon et al., 2018, Stevens et al., 2018, Young et al., 2018).

**Minor comments:**
Abstract:
1. Although it's mentioned in the title, I think it should be clarified also in the abstract that you are looking at instantaneous aerosol perturbations.
We removed the term "short-term" from the title, as we have moved the focus of the manuscript away from timescales. But we added a statement in the abstract, that the aerosol perturbations are instantaneous (page 1, line 3).
2. The sentence starting with "Motivated by ongoing sea ice retreat…" does not read very well. It's not clear what you contrast with what and that it is model simulations you are referring to.
We changed this sentence to "*Motivated by ongoing sea ice retreat, we performed all sensitivity simulations over open ocean and sea ice to investigate the effect of changing surface conditions.*" (page 1, lines 7-8).

Introduction:
3. Page 2, lines 4-6: Deep convective clouds are also mixed-phase.
Thank you, that is indeed true, we added that (page 2, lines 6-7).
4. Page 2, line 10: "… potentially causing a warming effect…" – why potentially? Isn't the LW (surface) effect always warming?
True, we deleted "potentially".
5. Page 2, line 15: "… eventually accelerating…" – why accelerating? Isn't it also possible that we could have a negative feedback from clouds?
That's true, especially during summer a higher cloud fraction might lead to a net cooling. We changed it to "*accelerating or slowing*" (page 2, line 16)
6. Page 2, lines 25-29: This sentence does not read very well.
We changed this sentence to "*These observed changes in cloud height were also observed during the Aerosol-Cloud Coupling And Climate Interactions in the Arctic (ACCACIA) campaign (Young et al., 2016). Besides, the authors reported fewer and larger cloud droplets as well as increased precipitation rates over the open ocean compared to over sea ice.*" (page 2, lines 28-31).
7. Page 2, line 33: Why would sea salt and dimethyl emissions dominate ship emissions? Do you mean that these (sea salt and DMS) emissions generally are larger than ship emissions?
According to Gilgen et al., 2018 the impact on CCN and hence on $N_{drop}$ of natural emissions from the ocean (such as sea salt and DMS) together with a changed future meteorology exceeds the impact that predicted ship emissions have on the cloud properties. Even with 10-fold ship emissions there was no significant impact on cloud properties.
However, for clarification we rewrote this section to "*An increased availability of cloud condensation nuclei (CCN) resulting from both, sea salt and dimethyl sulfide emissions from*

*the ocean and predicted ship emissions may lead to increased cloud formation and a net surface cooling during summer, as projected by global climate and earth system models (Gilgen et al., 2018, Stephenson et al., 2018)."* (page 3, lines 6-8).

Model description and setup:
8. Page 3, line 30: "km" should be "km$^2$".
We adjusted this to 20 km x 20 km.

Evaluation of background state:
9. Page 5, line 3: I would specify that "both control simulations" refer to the control simulations over open water and ice, respectively (and thereby define *ocean_control* and *ice_control*).
Specified (page 6, lines 5-6).
10. Page 5, line 9: "… lower in our model simulations". Lower than what? I assume you mean compared to observations?
Lower as compared to *observations*, we included that specification in the text (page 6, line 16).
11. Figure 1: Perhaps refer to table 2 for the simulations?
We included a reference to Table 1 (simulation overview) in the figure caption.
12. Table 2: "Cloud extend" should be "cloud extent".
This has been removed from the table (see reply to comment 3a).

Surface flux impact on cloud dynamics:
13. Figure 3: What time step is plotted?
After 3 h, we included that in the figure caption.
14. Page 6, line 5: "Cloud extend" should be "cloud extent".
This has been removed from the table (see reply to comment 3a).

Invariance of results across temperature regimes:
15. Page 13, line 10: I would suggest adding that the RH is kept constant, just as a clarification.
Added (page 17, line 3).

Consistent response independent of perturbation injection period:
16. Page 15, line 9: I would suggest adding "substantial" in between "no" and "change".
This section has been removed in the revised manuscript.

Discussion:
17. Page 16, line 15: What is tau?
We now consistently use the wording "cloud optical depth".
18. Page 17, line 4: Define WRF?
This sentence has been deleted in the revised manuscript (as we do not compare the aerosol accumulation to results by Solomon et al., 2018 any more).
19. Page 19, line 15: I suggest changing "resembling" to "providing" or something similar.
We changed to "providing" (page 22, line 16).

---

## Author Response (AR2)

**Response to Reviewer**

**Review**

In this study, the authors perform LES modelling of an Arctic mixed-phase cloud case, with the surface fluxes changed to represent either open ocean or a sea-ice-covered surface. They perform sensitivity studies to an instantaneous increase in either CCN or INP concentrations. The paper is generally well-written and well-organized, and the results are clearly presented. The paper merits publication, provided that the following issues are addressed.

Thank you very much for the additional review. We incorporated your suggestions within the revised manuscript. Please find our responses below, which we marked in red.

**General comments:**

In both the abstract and the conclusions section, the authors state that for the perturbed open ocean cases, "Increased ice and precipitation formation relax the liquid cloud properties back to their unperturbed range." However, the cloud droplet number concentration remains elevated, the cloud droplet radii remain reduced, and the precipitation rate remains elevated. The LWP in the perturbed simulations only reaches the range of the control simulation during the last couple of hours of the simulation. It is not at all obvious that this new cloud state with an LWP similar to that of the unperturbed case is stable. The authors should comment on whether they expect this to be a robust result for longer simulations, or whether a thicker cloud may re-form or if the cloud may even dissipate due to the increased precipitation rate in the perturbed simulations if the duration of the simulations was extended by a few more hours.

We agree that the question whether the perturbed cloud over the ocean would be stable beyond 20 h cannot conclusively be answered with our simulations. Nevertheless, CCN get lost through increased precipitation in the perturbed simulations after 12 h or through immersion freezing, such that we don't expect the cloud to thicken or re-form as the additional CCN are lost.

Also, we don't think that the perturbed cloud over the ocean starts to dissipate, as the averaged surface precipitation over the last two hours of the simulation is approximately similar in the *Control*, 500CCN and 1000CCN simulations ( $0.05\pm0.004 \text{ mm h}^{-1}$ ,  $0.06\pm0.003 \text{ mm h}^{-1}$ ,  $0.06\pm0.01 \text{ mm h}^{-1}$ , respectively) and most likely is not sufficient to dissolve the whole cloud.

Nevertheless, we also calculated the differences in terms of radiation between the *Control*, *500CCN* and *1000CCN* simulations over the last two hours and indeed the net surface LW radiation remains perturbed by  $1 \text{ W m}^{-2}$  and  $2.5 \text{ W m}^{-2}$  in the 500CCN and 1000CCN, respectively (with less outgoing LW radiation in the perturbed simulations). Similarly, the net surface SW radiation is decreased from 8.1 W m-2 in *Control* to 5.5 W m-2 in *500CCN* and to 4.5 W m-2 in *1000CCN*. This difference is smaller than the average over the whole simulated period (Table 3 and Table S1), however, the radiation still remains perturbed. We adjusted our arguments in the discussion that the cloud macrophysical properties (LWP) are efficiently buffered by precipitation and ice formation, but that the cloud microphysical as well as radiative properties remain changed in the perturbed simulations.

When discussing the change in the net surface LW, the authors do not mention whether the LW emission by the surface is different between the open ocean surface and the ice-covered surface. Presumably, the ocean surface is at a higher temperature than the ice-covered surface. This should be mentioned and discussed when contrasting the open ocean and sea-ice simulations, as all differences in the net surface LW between these simulations currently seem to be attributed only to differences in the cloud properties.

Thank you for bringing up this additional point. The surface emissivity is kept constant between the open ocean and sea ice surface. However, the ocean temperature is 273.15 K while the sea ice is slightly colder, at 270 K. As a result, the difference in surface temperature slightly changes the total LW emission by the surface. For the first hour of the simulation before the cloud started to form, the outgoing net surface LW radiation over the ocean is increased in magnitude as compared to sea ice by  $2.4\pm1.1$  W m-2 in the temporal/spatial mean. We agree that this does impact the total cloud radiative effect, but this is still below the simulated differences in net surface LW between the open ocean and sea ice cloud of 4.3 W m-2. Hence, we suggest the different sensible and latent heat fluxes over both surfaces rather than the surface LW emission being responsible for the cloud properties over the open ocean and sea ice.

We rewrote this paragraph on page 18 to: "In terms of radiative effects, the cloud over the open ocean and sea ice have different impacts on the net surface radiative balance. Note that the prescribed surface emissivity for ocean and sea ice is unchanged in both simulations. However, due to the 3 K warmer ocean, the LW surface emission is slightly increased over the open ocean and was quantified as  $2.4\pm1.1$  W m-2 (spatiotemporal average over the first cloud-free hour). Additionally, we find cloud base height to be the dominating factor determining the net surface LW radiative balance for clouds sufficiently optically thick in the LW spectrum (LW and SW radiation fluxes are defined to be positive downwards throughout our study). As the cloud over sea ice has a lower cloud base, the cloud re-emits LW radiation at warmer temperatures, which reduces the net surface LW cooling (Table 3). The net surface SW radiation is directly coupled to cloud optical depth and by around 4 W m-2 lower over the ocean, where the optically thicker clouds reflects incoming solar radiation more efficiently. Hence, we can extrapolate that during months with sufficient incoming solar radiation, clouds over the ocean might have a net zero to a cooling effect as compared to clouds over sea ice (as also found by Gilgen et al., 2018)."

Additionally, if the authors have the data available to estimate changes in net surface shortwave (SW) radiation between simulations, it would be helpful to include them where net surface LW is discussed. If negligible, a single sentence would be sufficient, if not, the authors should consider adding an additional row to Table 3. If the authors cannot estimate changes in net SW, they should very briefly discuss how the changes in net surface LW would be expected to compare to changes in net surface SW.

We calculated net surface SW (daytime only) and added a row in each of the tables. The net surface SW radiation is directly coupled to the cloud optical depth, such that the reflection of incoming SW radiation is increased for a thicker cloud and less SW radiation reaches the surface. For CCN perturbations the effect on net surface SW is rather strong and exceeds changes in net surface LW radiation (which is especially important for summertime clouds).

For INP perturbations the effect on SW radiation is rather small and smaller than the LW effect.

We included a more detailed discussion of the radiative properties including net surface SW radiation of the perturbed clouds in the discussion.

**Specific comments and technical corrections:**

p1, line 14: "once doubled as compared to the background concentration and once increased by a factor of 3" It would be clearer to use consistent phrasing here, e.g. "doubled [...] and quadrupled" or "increased by 100% [...] and by 300%"

Changed to: "INP perturbations were increased by 100% and 300% as compared to the background concentration [...]".

p3, line 3: "aerosol concentration" -> "aerosol concentrations"

Changed.

p3, line 6: "both, sea salt" -> "both sea salt"

Changed.

p3, line 19: "Turbulent mixing and de- and entrainment" In this case, it would be much clearer to write the whole word: "Turbulent mixing, detrainment and entrainment"

Changed to "Turbulent mixing, entrainment and detrainment".

p4, line 15: "ammoniumbisulfate" -> "ammonium bisulfate"

Changed.

p4, line 21: The authors should more explicitly state somewhere here that the background CCN concentration is fixed, not prognostic like the INP concentrations and the CCN perturbations.

To improve clarity, we added one sentence in the model description "The fixed background CCN ensure that sufficient CCN are available throughout the whole simulation for droplet activation" (page 4, lines 106-107). We also added "[...] an additional, fully prognostic mode of potential CCN or INPs was released [...]" in setup of perturbation experiments section (page 5, lines 135-136).

p4, line 23: "the CCN concentrations were chosen to match the observed  $N_{drop}$  over the ocean" Do the authors mean over ocean and sea ice? This value is greater than the observed value over the open ocean.

We chose the CCN concentrations to match the observed  $N_{drop}$  over the ocean. According to Young et al., 2016 the observed  $N_{drop}$  over the ocean was  $63\pm30$  cm-3. As we do not expect every potential CCN to activate, we chose the CCN concentrations to match the high end of the observed  $N_{drop}$ , such that we end up with similar  $N_{drop}$ .

We added a clarification to the model description "As we do not expect every CCN to activate, we initialized with a CCN concentration larger than the mean  $N_{drop}$  measured over the ocean. The initialized CCN concentration is still within the spread of the measured  $N_{drop}$  range though." (page 4, lines 116-118).

p4, line 28: "aeorsol" -> "aerosol"

Changed.

p5, line 17: "high latitude" -> "high-latitude"

Changed.

p7, line 9: please add "smaller" after "5 µm".

Added.

p7, line 10: "We relate these differences": The current phrasing seems to imply that the authors will discuss this further later in the text, but they do not. Would the authors care to comment on how the differences in observed temperature and relative humidity profiles over the sea ice and open ocean are related to the differences between the ice\_control case and observed cloud properties over sea ice?

We agree that an explanation was missing. We added a paragraph at the end of page 7, lines 176 ff.: "We relate these differences in cloud properties between our simulated and the observed MPC to the difference in the observed and simulated thermodynamic profiles: the drier boundary layer observed over sea ice suppresses cloud droplet growth. Moreover, the warmer and more turbulent boundary layer over the open ocean favors collision-coalescence of cloud droplets, leading to larger and fewer Ndrop over the open ocean. By choosing the same initial conditions for our open ocean and sea ice simulations, these processes are not equally represented."

p7, line 15: "domain wide" -> "domain-wide"

Changed.

p12, lines 3-4: This sentence is confusing. The Twomey effect describes an increase in cloud reflectivity as cloud droplet number increases and cloud droplet radii decrease, neglecting changes in liquid water content. However, the authors seem to be relating the decrease in radius to an increase in liquid water content. Please explain.

We agree this sentence is confusing. We rewrote this sentence to: "This decrease in radius is expected according to the Twomey effect (Twomey 1974). In addition, we also see a 20% increase in liquid water content through a delay of warm rain formation." (page 12, lines 223-224).

p12, lines 5-6: "a further increase in perturbation strength from 500 cm-3 to 1000 cm-3 does not induce an additional increase in LWP." According to Fig. 7, the maximum mean LWP increases from ~215 g m-3 to >230 g m-3. So this sentence seems to be incorrect.

We intended to state that the increase in LWP is small, considering a doubling of CCN. We clarified this sentence to: "[...] however, a further increase in perturbation strength from 500 cm-3 to 1000 cm-3 induces only a slight increase in LWP" (page 12, lines 225-226).

p13, line 20: Are the authors trying to relate the decreased variability in LWP to a more significant effect of the perturbation on LWP, despite the magnitude of the effect being similar? This sentence isn't very clear.

We want to point out that the spatial variability of the cloud over sea ice is reduced as compared to the cloud over open ocean. Hence, a lower concentration of perturbation CCN is needed to significantly perturb the LWP. We clarified this in the text: "However, the spatial variability of LWP is reduced over sea ice due to the more stratiform cloud deck. Therefore, smaller perturbations in LWP are considered outside the background variability in polluted simulations above sea ice. Indeed a CCN perturbation of 100 cm-3 is sufficient above sea ice, while a perturbation of 200 cm-3 is needed above the ocean to induce LWP perturbations outside the simulated background conditions." (page 13, lines 253-256).

p13, line 33: "(Fig. S5)" Do the authors mean to refer to Fig. S1a here?

We moved the reference to Fig. 7c and Fig. S5c behind "[...] where the strong and rapid increase in ice mass reduces the liquid-phase response" (page 13, line 267 to page 14, line 268) and replaced the reference to Fig S5 with reference to Fig. S1a on page 14, line 270.

p15, line 4: "double the background concentration" -> "an increase equal to the background concentration"

**Changed.**

p15, line 7: "400%" -> "300%". The IWP increases by 300% to attain a value that is 400% of its original value.

**Adjusted.**

p15, line 10: "Investigating" -> "After investigating"

**Changed.**

p15, line 11: "clouds perturbed" -> "clouds with perturbed"

**Changed.**

p15, line 11: "on the expense" -> "at the expense"

**Changed.**

p19, line 11: "Earth surface" -> "the Earth's surface"

**Changed.**

p20, line 8: In the open ocean case, the authors have repeatedly stated that after a CCN perturbation, the cloud relaxes to its initial state. Based on Fig. 9, this is not the case for INP

perturbations over open ocean. The statement that "the longer-term cloud response is more affected by CCN perturbations" therefore seems to be in error.

Yes, that is true, we removed the comparison.

Table 1: "simulation" -> "simulations"

Adjusted.

Table 2: Does the condition that cloud ice content  $q_i > 0.001 \text{ g m}^{-3}$  apply to all values listed in the table, or only  $N_{ice}$  and  $R_{ice}$ ?

The condition only applies for the respective liquid or ice cloud property (hence  $q_i > 0.001$  g m-3only applies to Nice and Rice), we clarified this in the table caption.

Fig. 4: Please move "(qc >0.01 g m-3)" after "in-cloud".

**Adjusted.**

Fig. 5: Please specify that it is the vertical range that is restricted to where immersion freezing occurs.

We added the specification "vertical".

Fig. 11: In the lower-left subplot, it appears that one of the cloud droplets has escaped from the cloud.

**This has been fixed.**

Fig. S2: Please re-label the y-axes to Ndrop and Rdrop for consistency with the text.

**Adjusted.**

Figs. S3, S4, and S5: Please adjust the captions from "all sensitivity simulations" to "all CCN perturbation simulations"

**Adjusted.**

Fig. S4: It is difficult to see the control line, because of the overlap with the other simulations. From hour 7 to hour 10, I do not know what the cloud top height for the control simulation is. I greatly appreciate the authors' consistency in colouring and line style across figures, but perhaps it would be best to increase the thickness of the line for the control simulation in this figure, to make it visible.

We increased the thickness of the black control line, such that it is visible also behind the colored lines.

[revised manuscript text omitted]
} \ ({ m cm}^{-3}$ ) | $N_{ice} \left( \mathbf{L}^{-1} \right)$ | $R_{drop}$ (µm) | $R_{ice}~(\mu{ m m})$ |
|-------------------------|--------------------|------------------------------|------------------------------------------|-----------------|-----------------------|
| observations ocean      | 0.24±0.13          | 63±30                        | $0.55{\pm}0.95$                          | 10              | -                     |
| ocean_control           | $0.11 {\pm} 0.08$  | 48±15                        | $0.27{\pm}0.20$                          | 6.5±1.7         | $15.5{\pm}2.0$        |
| Young et al. (2017) LES | 0.06               | 100                          | 0.34                                     | 10              | 30                    |
| observations sea ice    | $0.05 {\pm} 0.04$  | 110±36                       | $0.47{\pm}0.86$                          | 5               | -                     |
| ice_control             | $0.06{\pm}0.05$    | $40{\pm}18$                  | $0.08{\pm}0.05$                          | $5.8{\pm}1.8$   | $18.0{\pm}2.9$        |

Figure 2. Average (2-20 h)  $N_{drop}$  (solid lines) and the sum of all CCN tracers, i.e. background and perturbation mode, (dashed lines) in the a) *ocean\_control* simulations as well as most perturbed *1000CCN* simulation and b) *ice\_control* simulations as well as most perturbed *1000CCN* simulations.

**4 Surface flux impact on cloud dynamics**

185

The simulated effect of surface fluxes is illustrated in Fig. 3, showing a snapshot of the updraft velocities and LWP over ocean and sea ice after 3 h of simulation time. The different surface conditions lead to two different cloud regimes: over ocean, where surface fluxes are increased, the updrafts are higher, leading to cumulus towers detraining into the stratus deck and to a domain-wide shallow stratocumulus cloud structure. Within the shallow cumuli the LWP increases up to 300 g m-2, 4 times

higher than in the surrounding stratus layer. In contrast, over sea ice the updrafts are low and a spatially homogeneous stratus forms. The LWP of the stratus cloud remains below  $80 \text{ g m}^{-2}$ .